# Magnetization switching in polycrystalline Mn₃Sn thin film induced by self-generated spin-polarized current

Hang Xie [1], Xin Chen [1], Qi Zhang [1,2], Zhiqiang Mu[3], Xinhai Zhang[2], Binghai Yan [4] ✉ & Yihong Wu [1] ✉

Electrical manipulation of spins is essential to design state-of-the-art spintronic devices and commonly relies on the spin current injected from a second heavy-metal material. The fact that chiral antiferromagnets produce spin current inspires us to explore the magnetization switching of chiral spins using self-generated spin torque. Here, we demonstrate the electric switching of non-collinear antiferromagnetic state in Mn₃Sn by observing a crossover from conventional spin-orbit torque to the self-generated spin torque when increasing the MgO thickness in Ta/MgO/Mn₃Sn polycrystalline films. The spin current injection from the Ta layer can be controlled and even blocked by varying the MgO thickness, but the switching sustains even at a large MgO thickness. Furthermore, the switching polarity reverses when the MgO thickness exceeds around 3 nm, which cannot be explained by the spin-orbit torque scenario due to spin current injection from the Ta layer. Evident current-induced switching is also observed in MgO/Mn₃Sn and Ti/Mn₃Sn bilayers, where external injection of spin Hall current to Mn₃Sn is negligible. The inter-grain spin-transfer torque induced by spin-polarized current explains the experimental observations. Our findings provide an alternative pathway for electrical manipulation of non-collinear antiferromagnetic state without resorting to the conventional bilayer structure.

Recently Mn₃X (X = Sn, Ge, Ga, Rh, Ir, Pt) based non-collinear antiferromagnets (AFMs) have attracted significant attention[1–8] due to their large anomalous Hall[1,2,4,5], anomalous Nernst[6], and magneto-optical Kerr[8] effects and importantly, all these effects are accessible via a moderate magnetic field. In addition, these non-collinear AFMs also host a wide range of exotic phenomena from magnetic Weyl fermions[9,10], to ferroic ordering of cluster octupole moment[11], spin-polarized current[12,13], and magnetic spin Hall effect[14]. These exciting findings show that the chiral-spin structure and the resultant Berry curvature affect profoundly the charge and spin transport properties

and make the Mn₃X-based AFMs not only appealing for fundamental studies of the interplay between magnetism and topological electronic states[15,16] but also promising for AFM-based spintronics[17–23].

For practical applications, it is important to prepare thin films using commonly available deposition techniques such as sputtering and demonstrate the possibility of electrical manipulation of the non-collinear AFM state, both of which have been reported by several groups recently[24–32]. Notably, Tsai et al.[31] have demonstrated deterministic switching of the AFM state of Mn₃Sn polycrystalline films in Mn₃Sn/heavy metal (HM) bilayers with a bias field applied in the

[1]Department of Electrical and Computer Engineering, National University of Singapore, Singapore 117583, Singapore. [2]Department of Electrical and Electronic Engineering, Southern University of Science and Technology, Xueyuan Rd. 1088, Shenzhen 518055, China. [3]State Key Laboratory of Functional Materials for Informatics, Shanghai Institute of Microsystem and Information Technology, Chinese Academy of Sciences, Shanghai 200050, China. [4]Department of Condensed Matter Physics, Weizmann Institute of Science, Rehovot 7610001, Israel. ✉e-mail: binghai.yan@weizmann.ac.il; elewuyh@nus.edu.sg

current direction, and showed that the sign of Hall voltage change caused by the AFM state switching is determined by both the relative directions of the current and bias field and the sign of spin Hall angle of the HM layer. The polycrystalline film consists of crystal grains with different orientations, and the AFM state switching was found to occur mainly when the kagome plane is perpendicular to the current and parallel to the polarization direction of the spin current injected from the HM layer, which is also confirmed by Takeuchi et al.[32] in epitaxial Mn$_3$Sn/HM bilayers. In addition, Takeuchi et al.[32] have observed field-free chiral-spin rotation of non-collinear AFM state in the epitaxial bilayer, and found that the threshold current density for spin rotation is much smaller when the spin current polarization is perpendicular to the kagome plane as compared to the in-plane configuration. Both findings corroborate the spin-orbit torque (SOT) scenario, i.e., the injection of spin Hall current from the adjacent HM layer is responsible for the magnetization switching or chiral-spin rotation. The realization of spin manipulation in AFM/HM bilayers will certainly facilitate its integration with the ferromagnet (FM)-based spin-orbitronics as both employ the same bilayer structure. However, apart from externally injected spin current, the non-collinear spin structure can also produce transverse spin current and longitudinal spin-polarized currents inside the AFM itself[12–14]. These self-generated spin currents are equally important in electrical manipulation of the non-collinear AFM states, particularly in polycrystalline samples, but their roles are unrevealed in previous studies[31,32].

Here we demonstrate current-induced deterministic switching of AFM state in Ta/MgO/Mn$_3$Sn/MgO/Ta multilayers in which the thickness of MgO is varied from 0 to 6 nm to control spin current injection from the Ta layer. Clear current-induced switching is observed at all MgO thicknesses investigated, and interestingly, the switching polarity reverses when the MgO layer is sufficiently thick (>~3 nm). Our results demonstrate that there are two competing mechanisms causing the switching, one is the SOT from the spin current injected from the Ta layer and the other is the inter-grain spin-transfer torque (IGSTT) due to the spin current from neighboring grains. Current-induced switching is also observed in samples with MgO or Ti seed layer, which further confirms the existence of self-generated spin current in Mn$_3$Sn, in particular in the case of MgO seed layer, the top, and bottom interfaces are symmetrical. We argue that, unlike FMs, polycrystalline Mn$_3$Sn possesses sharp domain walls defined by the grain boundaries due to large magnetocrystalline anisotropy in the kagome plane, which makes the IGSTT an efficient mechanism for manipulating the AFM state of individual crystallites. We corroborate our argument with the magnetic properties of polycrystalline Mn$_3$Sn films and magneto-optical Kerr effect (MOKE) imaging of current-induced switching of the crystal grains. Furthermore, the removal of the thick HM layer that was used in previous studies significantly increases the anomalous Hall effect (AHE) signal by more than one order of magnitude, making it more appealing for energy-efficient device applications. Accurate control of grain size and its distribution may provide an effective knob to tune the switching characteristics that is more suitable for emerging applications such as neuromorphic computing.

## Results

### Structural and magnetic properties

The current-induced switching experiments are performed on sputter-deposited multilayers (Fig. 1a): Ta(2)/MgO($t_{MgO}$)/Mn$_3$Sn($t_{Mn_3Sn}$)/MgO($t_{MgO}$)/Ta(1.5) (the numbers inside the parentheses indicate the layer thickness in nm). All the samples are fabricated on Si/SiO$_2$ substrates. Here, $t_{MgO}$ and $t_{Mn_3Sn}$ denote the thickness of MgO and Mn$_3$Sn, respectively. The bottom Ta(2) and top Ta(1.5) layers are the seed and capping layers, respectively, and at the same time, the bottom Ta(2) also functions as a spin current injector. The bottom MgO layer with a thickness ranging from 0 to 6 nm is added to control the amount of spin current injection into the Mn$_3$Sn layer, where $t_{MgO} = 0$ corresponds to Ta/Mn$_3$Sn/Ta. To make the layer structure as symmetric as

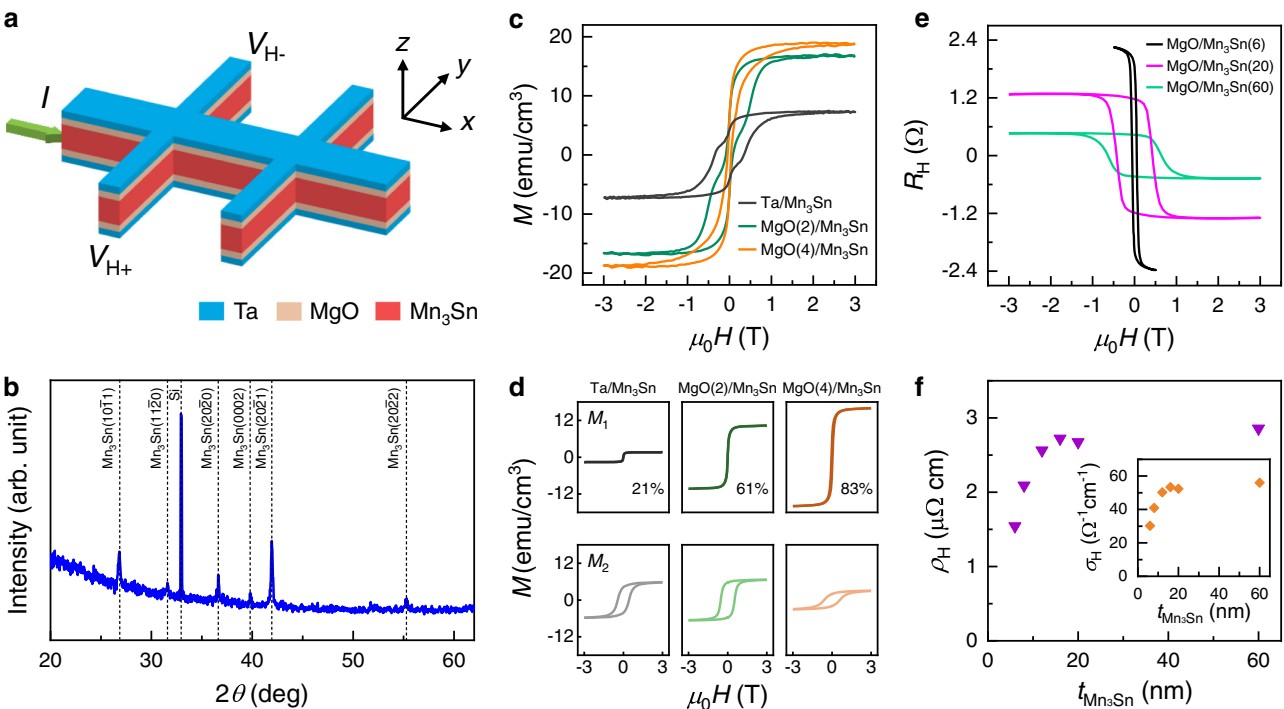

**Fig. 1 | Structural and magnetic properties. a** Schematic of the Hall bar device used in electrical measurement. Current is applied along *x*-direction, and Hall voltage is measured from two Hall probes $V_{H+}$ and $V_{H-}$ in *y*-direction. **b** XRD pattern of a Ta(2)/MgO(2)/Mn$_3$Sn(60) coupon film. **c** *M-H* loops for Ta/MgO($t_{MgO}$)/Mn$_3$Sn, $t_{MgO}$ = 0, 2, 4 nm at 300 K, with the magnetic field *H* along out-of-plane direction. **d** Decomposed sub-loops of the *M-H* loops in **c**. **e** Field dependence of Hall resistance $R_H$ for Ta/MgO/Mn$_3$Sn($t_{Mn_3Sn}$) with $t_{Mn_3Sn}$ = 6, 20, 60 nm at 300 K. **f** Extracted anomalous Hall resistivity $\rho_H$ as a function of $t_{Mn_3Sn}$. Inset: anomalous Hall conductivity $\sigma_H$ as a function of $t_{Mn_3Sn}$.

possible, we keep the thickness of the bottom and top MgO layers the same. The Ta(1.5) capping layer is presumably oxidized in ambience and can hardly contribute to spin current generation (hereafter MgO/Ta(1.5) is omitted for simplicity). Before we conduct the current-induced switching experiments, we first characterize the structural, magnetic, and transport properties of the multilayers.

Figure 1b shows the X-ray diffraction (XRD) pattern of a sputter-deposited $Mn_3Sn$ film with a thickness of 60 nm. As can be seen, multiple peaks corresponding to $(10\bar{1}1)$, $(11\bar{2}0)$, $(20\bar{2}0)$, $(0002)$, $(20\bar{2}1)$, and $(20\bar{2}2)$ planes of $D0_{19}$ phase $Mn_3Sn$ are observed, confirming the polycrystalline structure in sputter-deposited $Mn_3Sn$[24]. The peak positions remain the same for samples with different MgO thickness of 0, 2, and 4 nm (see Supplementary Information S1), suggesting that the insertion of MgO layer does not affect the crystalline structure of $Mn_3Sn$. We further characterize the magnetic properties of $Mn_3Sn$ with or without the MgO insertion layer using a superconducting quantum interference device (SQUID) magnetometer. Figure 1c shows the $M$-$H$ curves measured at room temperature with an out-of-plane magnetic field. As can be seen, both the saturation magnetization $M_s$ and the shape of $M$-$H$ loops change with the insertion of MgO layer. Although the $M_s$ increases from 7 emu/$cm^3$ at $t_{MgO} = 0$ to 16 emu/$cm^3$ at $t_{MgO} = $ 2 nm, and 19 emu/$cm^3$ at $t_{MgO} = 4$ nm, they are all on the same order of the saturation magnetization reported for polycrystalline $Mn_3Sn$ films at 300 K[28,33,34]. The $M$-$H$ loop apparently consists of two sub-loops with different saturation magnetization and coercivity. As detailed in Supplementary Information S2, the $M$-$H$ loop can be decomposed into two main components using the relation $M = \frac{2M_s}{\pi} \sum_{i=1}^{2} \alpha_i \text{atan}[\beta_i(H \pm H_{ci})]$. Here, $M_s$ is the total saturation magnetization, $\alpha_i$ is the weightage of individual component, $\beta_i$ indicates how responsive each component is to an applied field, and $H_{ci}$ is the coercivity. The $\pm$ sign indicates the backward and forward sweeping curves, respectively. Figure 1d shows the decomposed $M$-$H$ loops for the samples whose total $M$-$H$ loops are shown in Fig. 1c. Based on the XRD data and the fact that the easy axis of $Mn_3Sn$ is in the kagome plane, we can establish the following correlation: $M_1 - (11\bar{2}0)$ and $(20\bar{2}0)$, $M_2 - (20\bar{2}1)$, $(10\bar{1}1)$ and $(20\bar{2}2)$ planes. As can be seen from Fig. 1d, the MgO underlayer promotes the formation of $(11\bar{2}0)$ and $(20\bar{2}0)$ planes, both of which have their kagome plane perpendicular to the substrate. It should be noted that there should also be a small contribution from the $(0002)$ plane due to the out-of-kagome plane component, but it is difficult to separate it from the diamagnetic contributions from the substrate. The above results show that polycrystalline $Mn_3Sn$ films can be considered as consisting of magnetically weakly coupled crystal grains with different orientations. The grain boundary functions as an atomically sharp domain wall which makes the IGSTT more important in non-collinear AFM than in its ferromagnetic counterpart.

Next, we study the transport properties by patterning the $Mn_3Sn$ stacks into Hall bar devices (Methods for details). Figure 1e shows the anomalous Hall resistance $R_H$ of Ta/MgO/$Mn_3Sn$ with varying $Mn_3Sn$ thickness of 6, 20, 60 nm and a fixed MgO thickness of 2 nm. As can be seen, the $R_H$ increases with decreasing the $Mn_3Sn$ thickness due to both the increase of longitudinal resistance and enhancement of crystalline orientation alignment (Fig. 1e). The latter can be inferred from the decrease of coercivity when the thickness decreases. By fitting the longitudinal resistance as a function of $t_{Mn_3Sn}$, we obtain the resistivity of $Mn_3Sn$ and Ta as 225.9 and 680.6 $\mu\Omega$ cm, respectively (see Supplementary Information S3). Using these values, we can calculate the anomalous Hall resistivity ($\rho_H$) and conductivity ($\sigma_H$) as a function of $t_{Mn_3Sn}$, and the results are shown in Fig. 1f. As can be seen, although $\rho_H$ decreases with decreasing the thickness, it maintains a sizable value down to 6 nm below which the AHE becomes diminishingly small at room temperature. In contrast, the AHE for Ta/$Mn_3Sn$ disappears when $t_{Mn_3Sn}$ is below 12 nm (See Supplementary Information S3), presumably caused by the interdiffusion of Ta into $Mn_3Sn$. Therefore, although our original intention was to use the MgO to suppress the spin current flow

from the Ta layer to $Mn_3Sn$, it turned out that it also improves the magnetic properties, possibly due to the reduced interdiffusion at the Ta/$Mn_3Sn$ interface. It is worth noting that, although $Mn_3Sn$ is an AFM, the $R_H$ shown in Fig. 1e is comparable to that of typical HM/FM bilayers with a similar thickness, due to its large AHE induced by the Berry phase[4].

## Current-induced switching in $Mn_3Sn$

Current-induced switching measurements are performed for Ta/MgO/$Mn_3Sn$ with different $t_{MgO}$. Current pulses with a constant pulse width of 5 ms but varying amplitude are applied along the longitudinal direction, i.e., $x$-direction, of the Hall bar. In-between two adjacent writing pulses, the AFM state of $Mn_3Sn$ is read by current pulses with the same duration as that of the writing pulses but at a much smaller amplitude of 1 mA. Figure 2a shows the field-dependent Hall resistance of Ta/MgO/$Mn_3Sn$ with a fixed $Mn_3Sn$ thickness of 12 nm but varying $t_{MgO}$ from 0 to 6 nm. The extracted anomalous Hall resistivity $\rho_H$ is 0.28 $\mu\Omega$ cm at $t_{MgO} = 0$, but it increases quickly to 2.13 $\mu\Omega$ cm at $t_{MgO} = 1$ nm, and 2.56 $\mu\Omega$ cm at $t_{MgO} = 2$ nm, after which it maintains a constant value of -2.09 $\mu\Omega$ cm. Again, the large enhancement from $t_{MgO} = 0$ to 1 nm can be accounted for by the change in crystalline orientation induced by the MgO layer.

Figure 2b, c show the Hall resistance $R_H$ as a function of pulsed current amplitude $I$ in an applied longitudinal field $H_x$ of +600 Oe and −600 Oe, respectively, for six Ta/MgO/$Mn_3Sn$ samples with different $t_{MgO}$. We observe current-induced switching behavior for all the samples, which exhibit opposite switching polarity of $R_H$ upon reversing either the current or applied field direction. The switching of AFM state in Ta/$Mn_3Sn$ ($t_{MgO} = 0$) is consistent with previous reports on W/$Mn_3Sn$ and Ta/$Mn_3Sn$ based on the SOT scenario[31,32]. However, interestingly, the switching also occurs even with an MgO insertion layer between Ta and $Mn_3Sn$. In addition, it can be noticed that the switching polarity of $Mn_3Sn$ suddenly changes when $t_{MgO}$ increases to ~3 nm and beyond. Figure 2d shows the extracted switching current density in the Ta and $Mn_3Sn$ layer based on current distribution in each layer. The switching current density in the Ta layer is around $3.2 \times 10^6$ A $cm^{-2}$, comparable with those of previous reports, whereas the role of the spin current in the $Mn_3Sn$ layer will be discussed shortly.

The SR, i.e., the ratio between the current and field switched Hall resistance, is shown in Fig. 2e as a function of $t_{MgO}$. The change of sign at $t_{MgO} = 3$ nm is due to the change of switching polarity. We can see that the SR is positive and decreases monotonically from 61% to nearly zero at $t_{MgO} = 2.5$ nm (as comparison, the previously reported W/$Mn_3Sn$ device has a SR of ~25%)[31]. After the polarity reverses at around 3 nm, the SR becomes almost constant at ~14% till $t_{MgO} = 6$ nm. The solid-line in Fig. 2e is the fitted switching ratio (SR) according to SR $= -0.18 + 1.764e^{-0.885t_{MgO}}$, from which we can see that the SR consists of two components, one is negative with a constant amplitude and the other is an exponentially decaying contribution with a decay length of ~1.1 nm. The results point clearly to the fact that, in addition to the spin current from the Ta layer which should decrease exponentially with $t_{MgO}$, there must be another mechanism originated from the spin current or spin-polarized current inside the $Mn_3Sn$ layer, which is independent of the MgO thickness. Similar results are also observed for devices with different $Mn_3Sn$ thickness (see Supplementary Information S3). In addition, we have also confirmed from temperature-dependent measurement that the switching only occurs in the temperature region in which $Mn_3Sn$ is in the non-collinear AFM phase (Fig. 2f, Supplementary Information S4). The SR begins decreasing around 240 K, which is the transition temperature from the AFM phase to a spiral spin structure[24,35].

Furthermore, we performed current-induced switching in MgO(3)/$Mn_3Sn$(12) and Ti(2)/$Mn_3Sn$(12) (both are also covered with the MgO/Ta capping layers), where the Ta seed layer is replaced by MgO or Ti layer with negligible spin Hall angle. In both cases, external

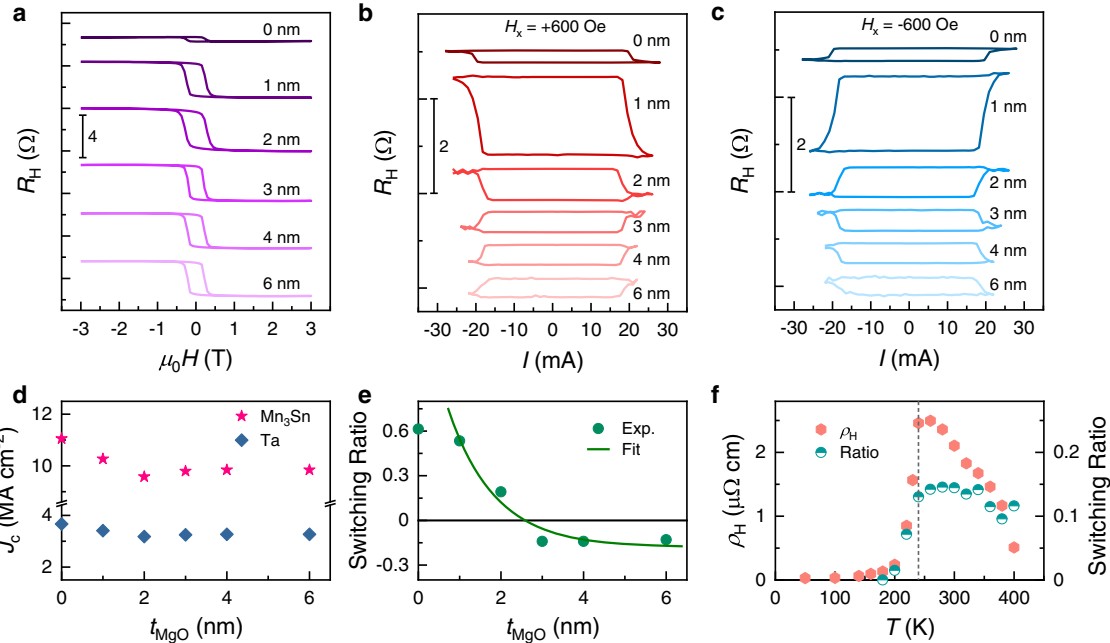

**Fig. 2 | AHE and current-induced switching in Mn₃Sn with varying MgO thickness. a** Field dependence of Hall resistance of Ta(2)/MgO($t_{MgO}$)/Mn₃Sn(12) at varying $t_{MgO}$ from 0 to 6 nm. **b, c** Current dependence of Hall resistance of Ta(2)/MgO($t_{MgO}$)/Mn₃Sn(12) at varying $t_{MgO}$ from 0 to 6 nm with an in-plane assistant field $H_x$ of +600 Oe and −600 Oe, respectively. **d, e** Extracted critical current density $J_c$ and switching ratio as a function of $t_{MgO}$. The solid-line in **e** is an exponential fitting. **f** Anomalous Hall resistivity $\rho_H$ and switching ratio of Ta(2)/MgO(2)/Mn₃Sn(12) as a function of temperature.

injection of spin Hall current to Mn₃Sn is negligible. As shown in Fig. 3a, d, both MgO(3)/Mn₃Sn(12) and Ti(2)/Mn₃Sn(12) show evident AHE with $\rho_H$ of 2.16 μΩ cm and 0.94 μΩ cm, respectively. The former is comparable with the AHE in Ta/MgO/Mn₃Sn while the latter is half the value. As shown in the current dependence of Hall resistance displayed in Fig. 3b, c, e, f, deterministic current-induced switching is also observed in both structures with an SR ~11% and 20%, respectively. It is interesting to note that the switching polarity of Ti/Mn₃Sn is the same as that of Ta/Mn₃Sn but opposite of Ta/MgO/Mn₃Sn with a thick MgO spacer and MgO/Mn₃Sn. Put the switching polarity aside (which will be discussed later), the results indeed suggest that it is possible to manipulate the AFM state of polycrystalline Mn₃Sn without an extra spin Hall layer but using the self-generated spin-polarized current. Besides, we also performed current-induced switching in Mn₃Sn directly deposited on Si/SiO₂ substrate (Fig. 3h, i). The much smaller remanence in its AHE loop (Fig. 3g) compared to other structures above indicates that grain size and its orientation in Si/SiO₂/Mn₃Sn can be quite different compared to other samples, resulting in negligible current-induced switching.

## MOKE imaging of current-induced switching

The low SR suggests that the switching of Mn₃Sn does not necessarily follow the two dominant mechanisms in SOT-based switching of HM/FM bilayers[36–41], i.e., coherent rotation (for small device)[36,37,40] or domain wall nucleation and propagation (for large device)[38–41]. In order to gain insight, we perform scanning MOKE imaging of the current-induced switching process. Here, we employ the polar mode MOKE to probe the magnetic state in Mn₃Sn (see Methods). A pulsed current with an amplitude larger than the threshold current and a pulse width of 5 ms is injected along the longitudinal direction of the Hall bar device. An in-plane field $H_x$ along the current direction is also applied to assist the switching. Figure 4a, b show the switching of Ta(2)/MgO(2)/Mn₃Sn(12) after the injection of a −25 mA pulsed current (upper panel) and 25 mA (lower panel) at $H_x$ of +460 Oe and −460 Oe, respectively. The MOKE image of the Hall bar changes from light gray to dark gray ($H_x$ = +460 Oe) and from dark gray to light gray

($H_x$ = −460 Oe) when the current changes from −25 mA to +25 mA, which indicates the switching of AFM state in Mn₃Sn. The light gray and dark gray images correspond to positive and negative anomalous Hall resistance[8]. It should be noted that it is difficult to estimate the SR from the MOKE image due to the limited spatial resolution of MOKE imaging. Consistent with the electrical measurement results, the reversal of switching polarity is also observed via MOKE imaging in samples with thick MgO layer, as can be seen from the MOKE images of Ta(2)/MgO(4)/Mn₃Sn(12) in Fig. 4c, d, respectively. In addition, we also perform the current-sweeping measurements. However, for both structures, we do not see clear domain wall nucleation and propagation from the MOKE images as reported for field-induced switching in single-crystal Mn₃Sn[8]. Instead, with increasing the current, the AFM state reverses gradually in localized areas across the sample (see Supplementary Information S5), resulting in graduate change of the MOKE image contrast. The results imply that the magnetic state reversal begins and ends with some of the crystallites instead of the entire sample, which explains why the SR is low. Furthermore, the switching becomes more non-uniform in the $t_{MgO}$ = 4 nm sample as compared to the $t_{MgO}$ = 2 nm sample (Fig. 4a, b), which is in good agreement with the electrical transport measurements.

## Discussion

The aforementioned results suggest that in addition to the external spin current from the Ta layer, there is another spin current source contributing to the spin dynamics of a particular grain in polycrystalline Mn₃Sn, which is from neighboring grains with misaligned spin sublattices. If we focus on a particular crystalline grain in Ta/MgO/Mn₃Sn, the dynamics of the magnetization vector $\mathbf{m}_\xi$ ($\xi$ = A, B, C) of each kagome sublattice can be described by the Landau-Lifshitz-Gilbert-Slonczewski (LLGS) equation

$$\frac{(1+\alpha^2)}{\gamma}\frac{\partial \mathbf{m}_\xi}{\partial t} = -\gamma \mathbf{m}_\xi \times \left[\mathbf{H}_{\xi,\text{eff}} - \alpha A_\parallel \sum_{i=1}^{n}\eta_i P_i I_{ci}\mathbf{s}_i\right] - \mathbf{m}_\xi$$
$$\times \left[\mathbf{m}_\xi \times \left[\alpha\mathbf{H}_{\xi,\text{eff}} + A_\parallel \sum_{i=1}^{n}\eta_i P_i I_{ci}\mathbf{s}_i\right]\right], \quad (1)$$

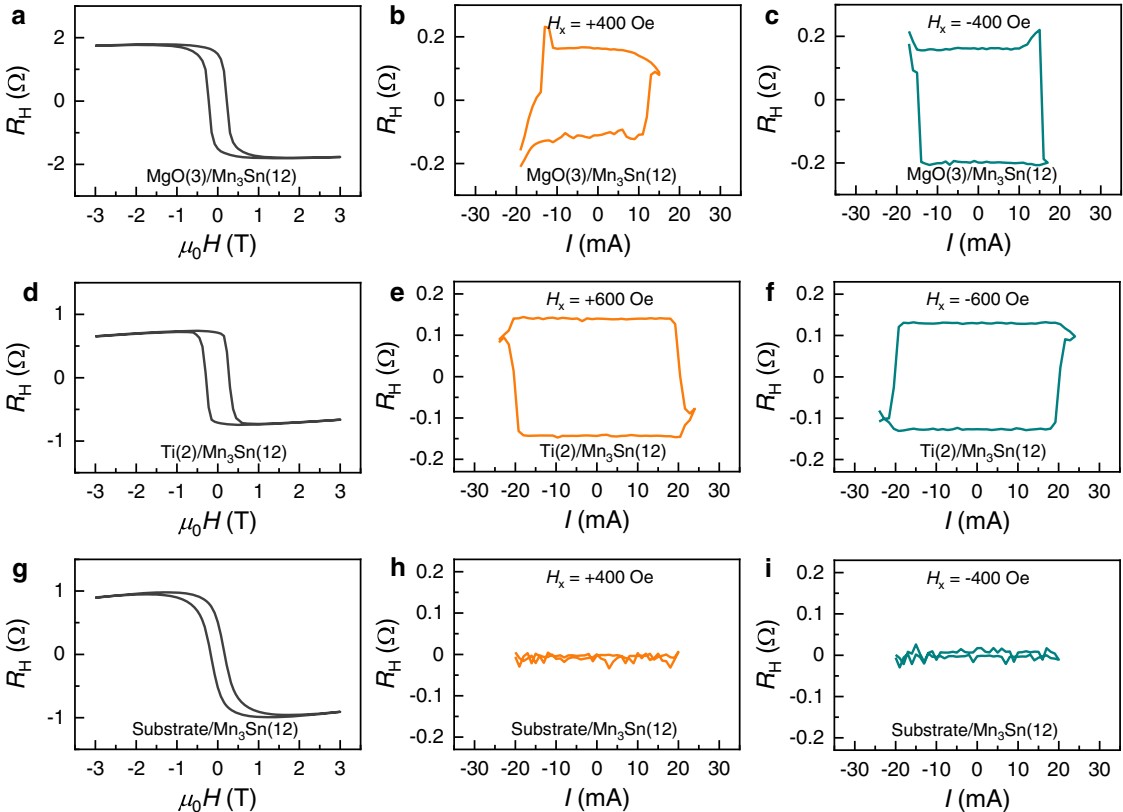

**Fig. 3 | AHE and current-induced switching in Mn₃Sn with MgO, Ti seed layer or directly grown on substrate. a, d, g** Field dependence of Hall resistance of MgO(3)/Mn₃Sn(12), Ti(2)/Mn₃Sn(12), and Substrate/Mn₃Sn(12), respectively. **b, e, h** Current dependence of Hall resistance of MgO(3)/Mn₃Sn(12), Ti(2)/ Mn₃Sn(12), and Substrate/Mn₃Sn(12) with a positive in-plane assistant field $H_x$, respectively. **c, f, i** Current dependence of Hall resistance of MgO(3)/Mn₃Sn(12), Ti(2)/Mn₃Sn(12), and Substrate/Mn₃Sn(12) with a negative $H_x$, respectively.

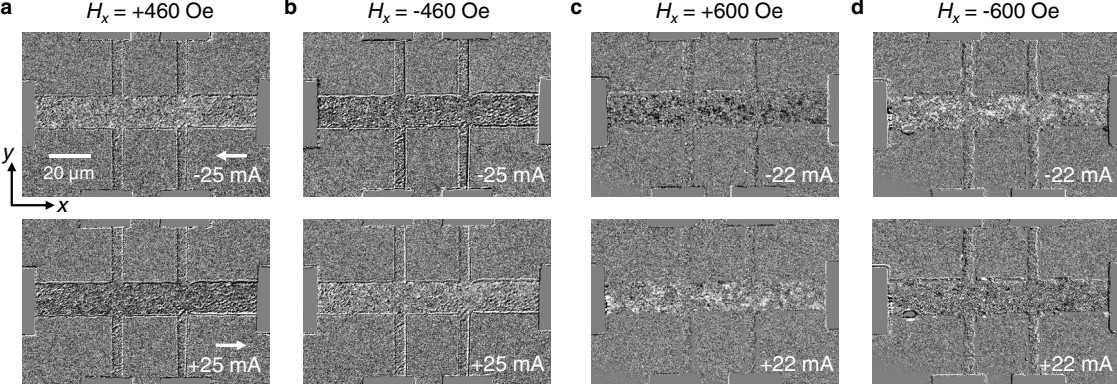

**Fig. 4 | MOKE images showing current-induced switching in Mn₃Sn. a, b** The images of Ta(2)/MgO(2)/Mn₃Sn(12) after application of a longitudinal current pulse with an amplitude of −25 mA (upper panel) and 25 mA (lower panel) in an longitudinal field $H_x$ of +460 Oe and −460 Oe, respectively. **c, d** The MOKE images of Ta(2)/MgO(4)/Mn₃Sn(12) after application of a longitudinal current with an amplitude of −22 mA (upper panel) and 22 mA (lower panel) in a longitudinal field $H_x$ of +600 Oe and −600 Oe, respectively.

where $A_{\parallel} = \frac{\hbar}{2eVM_s}$, $\gamma$ and $\alpha$ are the gyromagnetic ratio and Gilbert damping constant, $P_i$, $I_{ci}$, $s_i$, and $\eta_i$ are the spin polarization/spin current angle of Mn₃Sn or spin Hall angle ($\theta_{SH}$) of Ta, charge current, spin polarization direction, and spin injection efficiency of $i$th spin current source which includes both the Ta layer and neighboring Mn₃Sn grains, respectively, $M_s$ and $V$ are the magnetization and volume of the crystal grain to be switched, $\hbar$ is the reduced Planck constant, $e$ is the electron charge, and $\mathbf{H}_{\varepsilon, \text{eff}}$ is the effective field which can be calculated from the total magnetic energy density of the three sublattices[31,32,42–45]. Equation (1) is obtained by eliminating the time-dependent term at the right-hand side of the standard LLGS equation[45]. The dynamic spin

configuration of Mn₃Sn under external stimulus can be obtained by numerically solving the coupled LLGS equations. However, in order to gain a fundamental insight without resorting to micromagnetic modeling, we may examine qualitatively how the ground state of Mn₃Sn will be affected by the spin currents based on symmetry analysis[43,46]. In this regard, the magnetic state of Mn₃Sn may be represented by an average magnetization vector $\mathbf{m} = \frac{1}{3}(\mathbf{m}_A + \mathbf{m}_B + \mathbf{m}_C)$ and two staggered order parameters $\mathbf{n}_1 = \frac{1}{3\sqrt{2}}(\mathbf{m}_A + \mathbf{m}_B - 2\mathbf{m}_C)$ and $\mathbf{n}_2 = \frac{1}{\sqrt{6}}(-\mathbf{m}_A + \mathbf{m}_B)$. When the in-plane anisotropy is sufficiently large, the out-of-kagome plane component is very small, then $\mathbf{n}_1$ and $\mathbf{n}_2$ may be parameterized by $\varphi$ as $\mathbf{n}_1 = \frac{1}{\sqrt{2}}(\cos\varphi, \sin\varphi, 0)$ and

$\mathbf{n}_2 = \frac{1}{\sqrt{2}}(\sin\varphi, -\cos\varphi, 0)$, where $\varphi$ is the azimuthal angle from the easy axis direction of $\mathbf{m}_C$. For a single domain state without any external stimulus, the equilibrium states are given by $\varphi = 0$ and $\pi$, respectively. The corresponding magnetization vectors are $\mathbf{m} = (-1, 0, 0)$ and $\mathbf{m} = (1, 0, 0)$, respectively, with a magnitude of the order of $\frac{K}{J_0}M_s \approx 0.01M_s$, where $K$ is the anisotropy constant and $J_0$ is the homogeneous exchange coupling energy. There are three equivalent easy axis directions with the other two pointing at $\varphi = \pm\frac{2\pi}{3}$, respectively. When an external field $H$ is applied in the kagome plane, the magnetization will be aligned with the external field direction, when $H$ exceeds the coercivity which is on the order of a few hundreds of Oe.

With the approximated ground state, we now discuss qualitatively the effect of spin current by replacing $\mathbf{m}_\xi$ in Eq. (1) with the average moment $\mathbf{m}$. Without losing generality, we consider three types of structures as illustrated in Fig. 5a, namely, (i) HM/SC-Mn$_3$Sn, (ii) PC-Mn$_3$Sn, and (iii) HM/PC-Mn$_3$Sn. Here, SC and PC denote single-crystal and polycrystal, respectively. In case (i), we only need to consider the spin current injected from the HM layer with the spin polarization $\mathbf{s}_{HM} = (0, \pm1, 0)$ (hereafter, we ignore the $\pm$ sign as it is dependent on the sign of spin Hall angle of the HM layer and current direction), and therefore, $A_\parallel \sum_{i=1}^n \eta_i P_i I_{ci} \mathbf{s}_i = A_\parallel \eta_{HM} \theta_{SH} I_{c,HM} \mathbf{s}_{HM}$. As revealed by the previous studies[31,32,43,44], the effect of spin current on the AFM state depends on the direction of $\mathbf{s}_{HM}$ with respect to the kagome plane; binary switching occurs when $\mathbf{s}_{HM}$ is in the kogome plane, whereas spin rotation dominates when $\mathbf{s}_{HM}$ is perpendicular to the kagome plane. Based on Eq. (1), field-free switching will occur when the kagome plane is parallel to the $xy$-plane as there is always a finite projection of $\mathbf{s}_{HM}$ on one or more of the three easy axes. However, such a switching cannot be detected by the AHE in the present measurement configuration as the Hall voltage probes are along $y$ axis. The configuration which allows for both magnetization switching and AHE detection is when the kagome plane is parallel to the $yz$-plane. In this case, however, a longitudinal bias field $H_x$ is required to break the symmetry, same as the SOT-based switching in HM/FM bilayers[31,32]. When the kagome lattice is in the $zx$-plane, there will be no deterministic switching as $\mathbf{s}_{HM}$ is perpendicular to the kagome plane.

Next, we consider case (ii), which consists of only a single-layer polycrystalline film. In this case, $A_\parallel \sum_{i=1}^n \eta_i P_i I_{ci} \mathbf{s}_i$ only has the contributions from the neighboring grains (to facilitate discussion, hereafter we refer to the neighboring grains as polarizer and the grains to be switched as analyzer). For ultrathin films, we may assume that there is only a single grain in the thickness direction. Furthermore, the average effects of spin current transverse to the charge current should be largely canceled out based on symmetry considerations. Therefore, only longitudinal spin current or spin-polarized current is considered to possibly contribute to switching. For a collinear ferromagnet, the equilibrium polarization of spin-polarized current is generally parallel with the local magnetization direction. However, recent studies suggest that transverse polarization also exists[47]. For the present case, the longitudinal or spin-polarized current plays the dominant role. According to J. Železný, et al.[12], with the presence of spin-orbit interaction, the spin-polarized current emanating from the polarizer contains both a transverse and a longitudinal polarization. As revealed by XRD and magnetic measurements, the crystalline grains in polycrystalline Mn$_3$Sn may be divided into three groups based on their kagome plane orientation with respect to the substrate: (i) $(11\bar{2}0)$ and $(20\bar{2}0)$, (ii) $(10\bar{1}1)$, $(20\bar{2}2)$ and $(20\bar{2}1)$, and (iii) $(0002)$. Grains with $(0002)$ orientation do not contribute to the measured AHE signal and therefore, it cannot be analyzer. In general, when the charge current passes through the polarizer, spin current with both transverse and longitudinal polarization is generated. In the case of configurations shown in Fig. 5b, c, only spin current with transverse polarization plays a role in switching, because the longitudinal spin polarization is perpendicular to the kagome plane. Here, the primed coordinate system is

the local coordinate system attached to the polarizer such that $x'$ and $y'$ are always in the kagome plane, which is to be differentiated from the global coordinate system shown in Fig. 5a. In the configuration shown in Fig. 5b, an assistant field $H_x$ is used to align $\mathbf{m}$ of the polarizer along $x$-direction. In this case the spin current from the polarizer may be written as $j_s = j_{x'x'}^{x'} + j_{x'x'}^{y'}$ (note $x$ and $x'$ are in the same direction in this case)[12]. By convention, we use $j_{\beta\gamma}^\alpha$ to denote the spin current flowing in $\beta$ direction with a polarization along $\alpha$, induced by a charge current in $\gamma$ direction. When the spin-polarized current enters the analyzer, spins with transverse polarization $j_{x'x'}^{y'}$, will lead to magnetization switching due to the STT. The switching polarity should depend on the delicate balance of the spin-polarized current from neighboring grains. On the other hand, in the configuration shown in Fig. 5c, the polarizer and analyzer inter-change their roles, in which STT-based switching is also possible when an assistant field is applied in $y$ direction (See Supplementary Information S9). However, as the spin polarization in the two cases may be different in amplitude, the ratio of switched regions can differ significantly in the two cases. In actual samples, the polarizer and analyzer are not necessarily always perpendicular to each other as revealed by both XRD and SQUID results. Figure 5d–f show three possible configurations which can lead to both magnetization switching and detectable AHE. Unlike cases in Fig. 5b, c, it is difficult to decompose $j_s$ into spin current with longitudinal and transverse polarizations in the local coordinates as the kagome planes of the polarizer and analyzer do not form a right-angle. In these cases, depending on the exact kagome plane orientation of the polarizer and analyzer, spin currents with both transverse and longitudinal spin polarization may cause the switching of the magnetization. In the current measurement configuration, phenomenologically the AHE signal is proportional to the vertical component of the net magnetization in the kagome plane. Therefore, the configurations shown in Fig. 5d–f would lead to switching with a polarity opposite to that of Fig. 5b, c.

As discussed earlier, the texture of polycrystalline Mn$_3$Sn is well reflected in its $M$-$H$ loop. From the decomposition of $M$-$H$ curves as shown in Fig. 1d, we find that in the case of Ta/MgO($t_{MgO}$)/Mn$_3$Sn, the contribution from group (i) grains increases with increasing the MgO thickness, indicating that MgO seed layer facilitates the formation of group (i) grains and the switching in MgO/Mn$_3$Sn might be dominated by the configuration in Fig. 5b, c. In addition, as shown in Supplementary Information S2, the $M$-$H$ loop for Ti(2)/Mn$_3$Sn(60) is very similar to that of Ta/MgO(2)/Mn$_3$Sn, at which sign reversal has not occurred yet, i.e., the switching polarity is the same for Ta/Mn$_3$Sn, Ti/Mn$_3$Sn, and Ta/MgO(2)/Mn$_3$Sn. The XRD profiles and temperature-dependent AHE of Mn$_3$Sn in different structures further suggest that there are mostly group (ii) and (iii) grains in Ti/Mn$_3$Sn (Supplementary Information S1 and S7). Therefore, the switching configurations in Fig. 5d–f may become dominant in Ti/Mn$_3$Sn. At this stage, however, a quantitative analysis of the switching polarity would be difficult considering the random distribution of the crystalline orientation of the grains.

Lastly, case (iii) is a combination of case (i) and (ii), which represents most of the experimental configurations including the Ta/MgO/Mn$_3$Sn trilayers of present study when the MgO is thin. However, when the MgO is thick, we will have case (ii). The effects of the SOT (due to externally injected spin current) and STT (due to self-generated spin-polarized current) can either add up or partially cancel out, depending on the spin Hall angle of HM and crystallite orientations. In the present case, it seems that they cancel out partially at small $t_{MgO}$ due to opposite sign. This explains why the switching polarity reverses at large $t_{MgO}$ when the SOT effect is mostly suppressed.

As mentioned above, from MOKE images for Ta(2)/MgO(4)/Mn$_3$Sn(12) in Fig. 4c, d, we can see that the switched areas are discrete and more non-uniformly distributed than Ta(2)/MgO(2)/Mn$_3$Sn(12) case. This is consistent with the inter-grain STT switching scenario as the switching occurs only when a pair of neighboring grains have

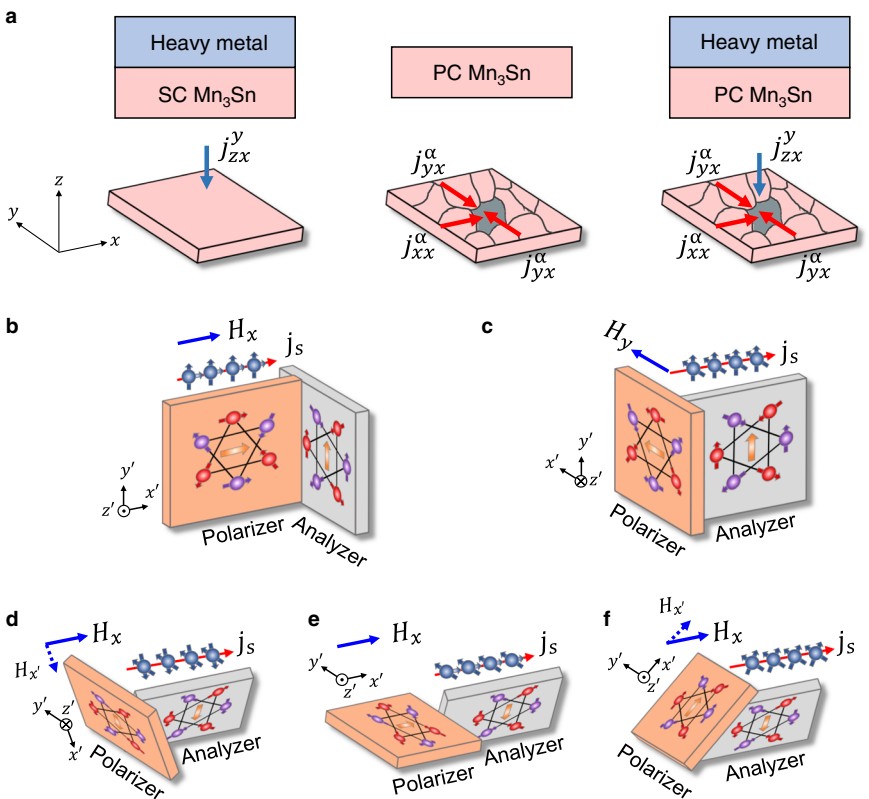

**Fig. 5 | Schematics of SOT and IGSTT-based switching. a** Three different structures for current-induced switching: heavy metal /single-crystalline (SC) $Mn_3Sn$, polycrystalline (PC) $Mn_3Sn$, and heavy metal /polycrystalline $Mn_3Sn$. **b**–**f** Schematic of IGSTT-based switching. The polarizer (left grain) denotes the grain which provides spin-polarized current, whereas the analyzer (right grain)

denotes the grain whose magnetic state is to be switched. $j^{\alpha}_{\beta\gamma}$: spin current flows in $\beta$ direction with a polarization along $\alpha$, induced by a charge current in $\gamma$ direction. The $x'y'z'$ coordinate system is the local coordinate system attached to the polarizer such that $x'y'$ are always in the kagome plane, and $z'$ is along the $c$-axis of the polarizer.

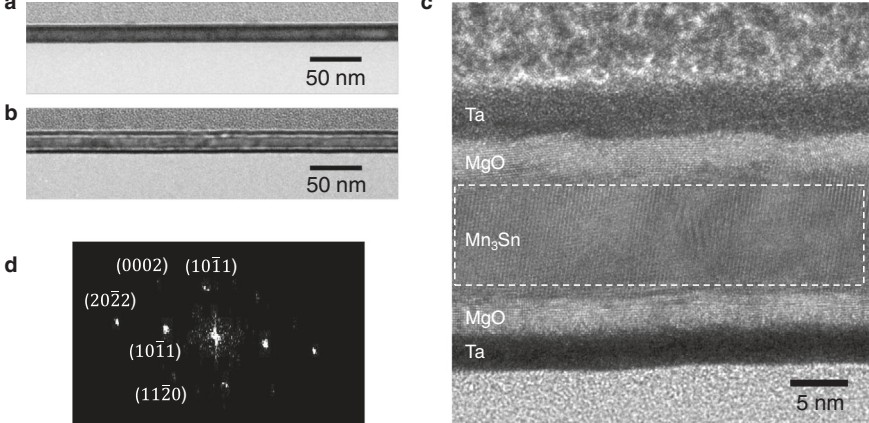

**Fig. 6 | TEM images of $Mn_3Sn$. a**, **b** Cross-sectional TEM images of Ta/$Mn_3Sn$ and Ta/MgO(4)/$Mn_3Sn$, respectively. **c** HR cross-sectional TEM image of Ta/MgO(4)/$Mn_3Sn$. **d** Electron diffraction pattern of the $Mn_3Sn$ layer in **c**.

specific configurations as one shown in Fig. 5b–f. However, it is not quite possible to determine the exact crystalline orientations of individual grains from the MOKE images due to limited spatial resolution of MOKE imaging. To support this scenario, we further examined the crystalline structure using transmission electron microscopy (TEM). From the cross-sectional TEM images in Fig. 6a, b, the $Mn_3Sn$ grain size ranges from 10 nm to 40 nm. Figure 6c displays the high-resolution (HR) cross-sectional TEM image for Ta/MgO(4)/$Mn_3Sn$, from which the structure for each layer is clearly seen. Despite the post-annealing at 450 °C, there is no apparent intermixing of $Mn_3Sn$ with Ta layer due to the MgO insertion layer, and the crystalline texture for $Mn_3Sn$ layer is

visible. By performing fast Fourier transform for the $Mn_3Sn$ layer in Fig. 6c, we obtained the corresponding electron diffraction pattern as shown in Fig. 6d. Based on the distances of the spots to the center, we further obtained the interplane spacings and the corresponding crystalline planes, as marked near each spot. The crystalline planes determined from TEM images is consistent with the XRD measurement. Multiple planes are detected in the small imaging area with width of ~37 nm, indicating the existence of grains with different crystalline planes in the lateral direction of the sample. In addition, the twin spots observed in Fig. 6d further confirm the grains existence and imply the existence of grain boundaries in the form of twin

boundaries[48–50]. A recent study also finds that there exist multiple grains in polycrystalline $Mn_3Sn$ and individual grain is like a domain with specific crystalline and magnetic orientation. The magnetic state in each grain switches independently due to the grain boundary[51]. A combination of in-situ TEM and spin-dependent scanning tunneling microscopy may be able to obtain more insights, which is out of the scope of this study.

The random distribution of crystal grain orientation and large size distribution may pose a challenge in applications where the device size has to be very small, e.g., non-volatile memory. In this regard, some of the techniques that have been developed for high-density magnetic recording media may be used to reduce the grain size and improve its size distribution. In addition, templated seed layer or grain boundary materials may be employed to control the crystalline orientations. On the other hand, nonuniformity in grain size distribution may also be exploited for emerging applications where gradual instead of abrupt switching is required, such as neuromorphic computing. One other limitation is incomplete switching, which is also reported in the other studies on HM/$Mn_3Sn$ systems[31,32,51]. From the current-induced switching in different structures, we find that the SR overall presents a positive correlation with the AHE remanence $R_{H(H=0)}/R_{AHE}$ (as shown in Supplementary Information S10) which is an indicator of the degree of texturing of the non-collinear AFM film. Therefore, further studies may be required to produce well-textured films with desirable grain orientation/size for current-induced switching. The active switching region may be reduced to contain only a few or ultimately two grains, thereby allowing to gain further insights into the switching mechanism and enhance the switching efficiency.

Before we conclude, it is worth mentioning that the proposed mechanism is different from previous reports on magnetization switching of single-layer FM or AFM which treat the single-layer largely homogeneous in terms of the magnetic order parameter. The mechanisms proposed in these early works include spin current with spin polarization transverse to the magnetization[52,53], AHE-induced surface spins with spin polarization rotation in FM[54], asymmetric absorption of internally generated spin Hall current in ferrimagnet[55], and current-induced internal staggered SOT due to local inversion asymmetry of spin sublattices in collinear AFM[17,19,56]. However, all these mechanisms cannot account for the results obtained in this study. In addition, the interface-related effect such as Rashba effect is not likely to play an important role here, as in samples of Ta/MgO/$Mn_3Sn$/MgO/Ta and MgO/$Mn_3Sn$/MgO/Ta, we always kept the bottom and upper MgO layer the same thickness to minimize structural asymmetry. If there is any significant Rashba effect between MgO and $Mn_3Sn$ interface, the effects from upper and lower interfaces will be largely canceled out. The interface-related effect also fails to account for the negligible switching in substrate/$Mn_3Sn$/MgO/Ta, as structural asymmetry is more prominent in this structure. Furthermore, a previous study has excluded the interface effect in current-induced switching of $Mn_3Sn$ by inserting a Cu layer between $Mn_3Sn$ and heavy metal layer[57]. Previous studies also suggest negligible charge-spin conversion at Ti/ferromagnetic interface[58,59]. In addition, as shown in Supplementary Information S8, the same switching polarity and similar level of SR in Ti/$Mn_3Sn$/Ti symmetric structure with Ti/$Mn_3Sn$ further attests that the switching is not induced by interface effect. After the submission of this manuscript, several theoretical and experimental works have been published, which support the spin current generation and grain-based switching mechanism discussed in this study[13,51,60].

In conclusion, we have demonstrated that current-induced AFM state switching is possible in single-layer $Mn_3Sn$ polycrystalline films via the inter-grain spin-transfer torque. As it does not require a heavy metal layer, the current shunting effect is significantly suppressed, which leads to a large anomalous Hall signal. Our results shed light on the crucial role of IGSTT in current-driven dynamics of polycrystalline non-collinear AFMs, which reveals the presence of self-generated spin

current in non-collinear AFM. The IGSTT may be used to realize devices such as inter-grain spin valve and tunnel junctions. We shall point out that the IGSTT has been studied before in a different context as a possible mechanism in magnetization reversal of heat-assisted magnetic recording, in which the magnetic grains are separated by oxide and the spin-polarized current is generated by heat gradient[61]. At last, it is worth mentioning that we do not completely rule out other possible mechanisms such as the presence of hidden structural asymmetry that is not apparent in the as-deposit layer structure. In addition, the joule heating effect has also been reported recently to play a crucial role in the switching behavior of $Mn_3Sn$[62,63]. We hope all these combined may stimulate more investigations in the current-induced switching of non-collinear antiferromagnets.

## Methods
### Sample preparation
The stacks, except the Ti layer, were deposited on Si/$SiO_2$ (300 nm) substrates using magnetron sputtering with a base and working pressure of $<1 \times 10^{-8}$ and $3 \times 10^{-3}$ Torr, respectively. The Ti layer was deposited by e-beam evaporation in the same system without breaking the vacuum. Standard photolithography and lift-off techniques were used to fabricate the Hall bars. The Hall bar length, width, spacing between voltage electrodes, and width of voltage electrode are 120 µm, 15 µm, 30 µm, and 5 µm, respectively. The Mircotech laser-writer system with a 405 nm laser was used to directly expose the photoresist (Mircoposit S1805), after which it was developed in MF319 to form the Hall bar pattern. After film deposition, the photoresist was removed by a mixture of Remover PG and acetone to complete the Hall bar fabrication. The processes of photolithography and lift-off were repeated to form electrodes and contact pads with the layers of Ta(5)/Cu(150)/Pt(10) for Hall bars. After the device fabrication and deposition, the devices were annealed at 450 °C for an hour in vacuum.

### Structural and magnetic characterizations
XRD measurements were performed in a Rigaku X-ray diffractometer. A Cu-Kα X-ray was used in the diffractometer with a wavelength of 1.541 Å. The cross-sectional transmission electron microscopy imaging was performed using a JEOL JEM 2100F. TEM specimen was prepared using focused ion beam milling. The magnetic properties were characterized using Quantum Design MPMS3, with the resolution of $<1 \times 10^{-8}$ emu.

### Electrical measurements
The electrical measurements were performed in the Quantum Design VersaLab PPMS with a sample rotator. The ac/dc current is applied by the Keithley 6221 current source. The longitudinal or Hall voltage is measured by the Keithley 2182 nanovoltmeter.

### MOKE imaging
The MOKE imaging of out-of-plane magnetic domain was performed in polar geometry in a wide-field Kerr microscopy from Evico Magnetics at room temperature. The in-plane magnetic field was generated by a pair of electromagnets. To enhance the contrast, the non-magnetic background was firstly subtracted. Afterwards, 10 repeated current pulses with amplitude larger than threshold current were injected to the long axis of Hall bar device. For the current-sweeping measurement, a large negative current pulse was applied first to initialize the magnetic state. Then, positive current pulses with increasing amplitude from zero were injected. Same current range was used as the electrical measurement. Images were taken after each pulsed current injection.

## Data availability
All the data related to this work are present in the main text and Supplementary Information. The data sets that support the findings of this work are available from the corresponding author upon reasonable request.

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

## Acknowledgements

Y.W. acknowledges the funding support by the Ministry of Education, Singapore under its Tier 2 Grants (Grant No. MOE2017-T2-2-011 and MOE2018-T2-1-076). B.Y. acknowledges financial support by the European Research Council (ERC Consolidator Grant, No. 815869) and the Israel Science Foundation (ISF No. 1251/19, No. 3520/20, and No. 2932/21). We thank professor Shufeng Zhang, Jianpeng Liu, and Guoping Zhao for helpful discussions.

## Author contributions
Y.W. designed and supervised the project. H.X. and Y.W. designed the experiments. H.X. conducted most of the experiments. X.C. helped with sample fabrication and XRD measurement. Q.Z. helped with MOKE imaging. Z.M. performed TEM imaging. H.X., B.Y., and Y.W. analyzed the data. All authors discussed the results. H.X., B.Y., and Y.W. wrote the manuscript and all the authors contributed to the final version of the manuscript.

## Competing interests
The authors declare no competing interests.
