## [Peer Review File · Nature Communications]

Reviewers' Comments:

Reviewer #1:

Remarks to the Author:

This article investigates the self-generated spin-transfer torque and the resultant switching of polycrystalline Mn₃Sn. The results in this work are different from the spin-orbit-torque-induced magnetic switching in previous works (such as Ref. 31 and 32). Thus this work could be considered for the publication in Nat. Commun. But there are a few questions about the inter-grain spin-transfer torque mentioned in this article.

(1) To prove the inter-grain spin-transfer torque in noncollinear AFM, the most direct method I think is to sputter Mn₃Sn on the Si substrate and see whether there exists switching. Why does the authors choose the Ta/MgO/Mn₃Sn and Ta/Ti/Mn₃Sn system rather than Si/Mn₃Sn system? Some additional experiments with more pure structure are encouraged.

(2) The inter-grain spin-transfer torque requires the spin current to flow across different grains and the magnetic switching behavior only occur in certain grains, which is also shown in the MOKE experiments in Fig.3. During the sputtering process, the sizes and distributions of the grains are unable to be uniform. It will cause inhomogeneity when applying noncollinear AFM to high-density devices which is the main driving force for the investigation of noncollinear AFM. How to solve the inhomogeneity problem?

(3) How to control the switching polarity in the polycrystalline situation? For example, the number of grains which generate +y spin polarization and that of grains generating -y spin polarization may be equal, or the former is larger than the latter sometimes while is smaller in other times.

(4) The assistant field is comparatively large. Is there any critical assistant field for the present Mn₃Sn. In general, the assistant field is used to break the symmetry from the macrospin viewpoint or conquer the DMI field from the multi-domain viewpoint. What is the function of the assistant field here?

(5) Whether the switching ratio is seriously reduced by polycrystalline feature with variable Kagome planes? Whether fully switching (when the thermal effect is not taken into account) can be realized for the SOT switching of Mn₃Sn?

(6) The SOT switching was realized in ferromagnetic single layer (Nature Nanotechnol. 14, 819 (2019), Phys. Rev. B 101, 214418 (2020), Adv. Funct. Mater. 2020, 2005201) and antiferromagnetic single layer (Science 351, 587 (2016), Nat. Commun. 9, 348 (2018), Nat. Mater. 18, 931 (2019)). This point should be mentioned in this work. And the origin for the SOT effect in different magnetic single layer should be compared.

Reviewer #2:

Remarks to the Author:

This work, entitled "Magnetization switching in polycrystalline Mn₃Sn thin film induced by self-generated spin-polarized current," demonstrates an experimental study on the subject, and the authors attribute the observed phenomenon to the inter-grain spin-transfer torque (STT) within the Mn₃Sn film.

The topic is timely and is of current interest of community. But I don't think the data displayed in the manuscript is convincingly supporting the authors' conclusion on the underlying mechanism of the observed switching.

The switching polarity is dependent on the normal metal (NM) layer, i.e., whether it is MgO or Ti, attached to Mn₃Sn. This may be indicating that the NM/Mn₃Sn interface is playing the key role, instead of the self-generated torque in Mn₃Sn. For example, whereas the systems in the present work may not be exactly known for exhibiting a large Rashba spin-orbit coupling, it is symmetrically allowed to appear here and it would also lead to the magnetization switching via the corresponding spin-orbit torques. The authors are not discussing this possibility. Some more systematic study is required to exclude possible influences of the interface.

An electric current passing through a grain (Polarizer) contributes to the switching of a neighboring grain (Analyzer) only when the specific relative crystalline orientations between the two grains are realized (there are two such configurations out of the twenty-four possible ones, as discussed by

the authors). And in any case, Polarizer grains themselves do not change their magnetizations under the current. So, there must be way more regions in a sample that are not affected by the current than those change their magnetizations in a way that the change is reflected in the MOKE signal. Considering these things, I wouldn't expect the proposed inter-grain STT leads to such an appreciable contrast in the MOKE signal as in Fig. 3. Can the authors make some comment on this point?

Also, if it is the case that the inter-grain STT indeed plays the essential role in the switching, the authors should be able to get some insights into the crystalline orientations of each grain contained in the sample by comparing the MOKE results before and after the current pulse applications, and discuss if the current-induced changes in the grains are consistent with the inter-grain-STT scenario.

As several groups have already reported experimental demonstrations of current-induced switching/rotation of the magnetizations in Mn₃Sn thin films, the (almost only) major advance of this work is the claimed elucidation of the new physical switching mechanism, that is, the inter-grain STT in polycrystalline thin films. Therefore, I believe that the authors must provide more clear-cut/direct experimental evidence that the inter-grain STT is indeed playing the dominant role in the observed switching.

In my eyes, the manuscript can be published somewhere else as there do not seem to be major problems in the experimental procedures/methods employed. But I don't see a reason that it should be published in a high-impact journal such as Nature Communications, unless the authors can address all the above-mentioned issues.

We would like to thank the reviewers for the detailed review of the manuscript and insightful comments. Below we include a summary of changes in the revision and point-to-point responses to reviewers' comments:

A) Summary of changes (highlighted in red in the revised manuscript)

- 1) Added current-induced switching results of MgO/Mn₃Sn and Substrate/Mn₃Sn.
- 2) Discussed the nonuniformity problem of grain size and distribution when the device size becomes smaller.
- 3) Elaborated the inter-grain STT induced switching mechanism and explained the switching polarity difference of Ti/Mn₃Sn and MgO/Mn₃Sn.
- 4) Added more discussion about the in-plane assistant field role in switching.
- 5) Discussed the incomplete switching of Mn₃Sn.
- 6) Summarized the SOT effect in single layer FM or AFM in previous studies.
- 7) Added experiment and discussion to rule out the interface effect contribution to current-induced switching.
- 8) Discussed the limitation of MOKE imaging resolution and the difficulty in extracting the switching ratio and crystalline orientation information from MOKE image.
- 9) Added TEM imaging results of Mn₃Sn to show the thin film structure and existence of grains in Mn₃Sn.

B) Response to comments from Reviewer #1

Comment 1: “To prove the inter-grain spin-transfer torque in noncollinear AFM, the most direct method I think is to sputter Mn₃Sn on the Si substrate and see whether there exists switching. Why does the authors choose the Ta/MgO/Mn₃Sn and Ta/Ti/Mn₃Sn system rather than Si/Mn₃Sn system? Some additional experiments with more pure structure are encouraged.”

Response: We would like to thank the reviewer for the thoughtful comment. First at all, we would like to clarify that all the samples are deposited on Si/SiO₂ substrate including the Ti/Mn₃Sn sample. We have added remarks in the revised manuscript. The Ti/Mn₃Sn sample does not have a Ta underlayer. The reasons for choosing the Ta/MgO/Mn₃Sn and Ti/Mn₃Sn structures in the study are as follows. In the case of Ta/MgO/Mn₃Sn, Ta serves as both a seed layer and a spin current generator, and the MgO layer facilitates the control of the amount of spin current that will be injected into Mn₃Sn via varying thickness. As for Ti/Mn₃Sn, the Ti layer is known to be a good seed layer and at

the same time it has negligible spin current generation. Therefore, we believe the combination of these two groups of samples helps to discriminate the roles of externally generated (from Ta) and self-induced (from neighbouring Mn₃Sn grains) spin current in current-induced switching of Mn₃Sn thin films. We didn't use Mn₃Sn directly deposited on Si/SiO₂ substrate as the control sample because these films are typically not well textured, i.e., quite different from the texture of Ta/MgO/Mn₃Sn, which will make it difficult to have an apple-to-apple comparison between the two.

Taking the reviewer's suggestion, we have deposited samples with the Si/SiO₂/Mn₃Sn(12) and Si/SiO₂/MgO/Mn₃Sn(12) structure, and further performed current-induced switching measurements. In both structures, we use top MgO/Ta protection layer to prevent Mn₃Sn from oxidation. Figure 1a and 1d below show the AHE loops for Si/SiO₂/Mn₃Sn(12) and Si/SiO₂/MgO/Mn₃Sn(12), respectively (hereafter, "Fig." refers to figures in this rebuttal unless specified otherwise). We can find that, compared to Si/SiO₂/MgO/Mn₃Sn(12), the AHE loop of Mn₃Sn directly deposited on Si/SiO₂ has a much smaller remanence ($\frac{R_{H(H=0)}}{R_{AHE}} \approx 52\%$) and lower anomalous Hall resistance R_{AHE} , indicating either poor texturing or small grain size, or mixture of non-crystalline phase in Si/SiO₂/Mn₃Sn(12), e.g., co-existence of Mn₂Sn (T. Ikeda, et. al., Appl. Phys. Lett. 113, 222405 (2018)). This may explain the negligible current-induced switching observed in Si/SiO₂/Mn₃Sn(12) as shown in Fig. 1b-c. A well textured Mn₃Sn films with large grain size is crucial for obtaining both large AHE and achieving current-induced switching based on the proposed inter-grain spin-torque transfer (IG-STT) mechanism. This is because the AHE in Mn₃Sn depends on the current and field directions relative to the crystal directions and the IG-STT only occurs across grain boundaries with specific grain orientations. In addition, Ta and Ti are often used as seed layers both to increase the adhesion and to enhance the texture of subsequently deposited thin films including ferromagnets and antiferromagnets. Therefore, it is not surprising that Mn₃Sn deposited on Ta or Ti seed layer would exhibit different structural and magnetic properties. From the results of Si/SiO₂/MgO/Mn₃Sn(12) shown in Fig. 1d-f, we can see that a thin MgO seed layer can also help to improve the crystallinity and texture of Mn₃Sn, leading to an AHE remanence twice of that in Si/SiO₂/Mn₃Sn. Importantly, evident deterministic current-induced switching is observed in Si/SiO₂/MgO/Mn₃Sn(12); its switching ratio also improves to around 11%, slightly smaller than the Si/SiO₂/Ta/MgO/Mn₃Sn(12) sample with $t_{MgO} \geq 3$ nm. The switching polarity is also the same as the latter. Due to the absence of heavy metal layer in Si/SiO₂/MgO/Mn₃Sn, the observed current-induced switching can only result from the self-generated spin-polarized current in Mn₃Sn. We believe the switching results of Si/SiO₂/MgO/Mn₃Sn are convincing enough to prove the inter-grain spin-transfer torque in noncollinear AFM. The interface effect is presumably negligible as the top interface is also

Mn₃Sn/MgO. Should the interface effect be dominant, it would be more obvious in SiO₂/Mn₃Sn/MgO than in MgO/Mn₃Sn/MgO, which is not the case based on experimental results.

The other point which we would like to stress is that we did not use single crystal or epitaxial Mn₃Sn because the focus of this work is to study the IG-STT as an alternative switching mechanism for current-induced switching of polycrystalline non-collinear AFM films. As has been discussed in the manuscript, single crystal Mn₃Sn can only be switched by spin current from an adjacent HM layer, while for polycrystalline Mn₃Sn, the self-generated spin current from the neighboring crystalline grains contribute to spin dynamics in Mn₃Sn as well. Our work provides strong support for theoretical proposals that current-induced switching can be realized in magnetic tunnel junctions made by non-collinear AFM (AFM (J. Železný, et. al., Phys. Rev. Lett. 119, 187204 (2017); D. Go et al., arXiv:2201.11476; J. Dong et al., Phys. Rev. Lett. 128, 197201 (2022)).

FIG. 1. **a, d**, Field dependence of Hall resistance of Si/SiO₂/Mn₃Sn(12) and Si/SiO₂/MgO(3)/Mn₃Sn(12), respectively. **b-c**, Current dependence of Hall resistance of Si/SiO₂/Mn₃Sn(12) with an in-plane assistant field H_x of +400 Oe and -400 Oe, respectively. **e-f**, Current dependence of Hall resistance of Si/SiO₂/MgO(3)/Mn₃Sn(12) with an in-plane assistant field H_x of +400 Oe and -400 Oe, respectively.

As a response to the reviewer's comment, we have made the following change:

a) Page 1, line 16-18:

Original: "Evident current-induced switching is also observed in Ti/Mn₃Sn bilayer, where spin Hall current from Ti is negligible."

Revised: "Evident current-induced switching is also observed in MgO/Mn₃Sn and Ti/Mn₃Sn bilayers, where external injection of spin Hall current to Mn₃Sn is negligible."

b) Page 3, line 75-78:

Original: “By replacing Ta with Ti seed layer, we further confirm the existence of self-generated spin current in Mn₃Sn.”

Revised: “Current-induced switching is also observed in samples with MgO or Ti seed layer, which further confirms the existence of self-generated spin current in Mn₃Sn, in particular in the case of MgO seed layer, the top and bottom interfaces are symmetrical.”

c) Page 4, line 90-91:

Added: “All the samples are fabricated on Si/SiO₂ substrates.”

d) Page 10, line 213-223:

Original: “Furthermore, we performed current-induced switching in Ti(2)/Mn₃Sn(12), where Ta seed layer is replaced by Ti with negligible spin Hall angle. As shown in Fig. 2g, with the Ti seed layer, the Mn₃Sn still shows evident AHE with ρ_H of 0.94 $\mu\Omega$ cm, which is more than three times of the value in Ta/Mn₃Sn but half the value in Ta/MgO/Mn₃Sn. As shown in the current dependence of Hall resistance displayed in Fig. 2h-i, deterministic current-induced switching is also observed in Ti/Mn₃Sn with a switching ratio around 20%.”

Revised: “Furthermore, we performed current-induced switching in MgO(3)/Mn₃Sn(12) and Ti(2)/Mn₃Sn(12) (both are also covered with MgO/Ta capping layers), where the Ta seed layer is replaced by MgO or Ti layer with negligible spin Hall angle. In both cases, external injection of spin Hall current to Mn₃Sn is negligible. As shown in Fig. 3a and 3d, both MgO(3)/Mn₃Sn(12) and Ti(2)/Mn₃Sn(12) show evident AHE with ρ_H of 2.16 $\mu\Omega$ cm and 0.94 $\mu\Omega$ cm, respectively. The former is comparable with the AHE in Ta/MgO/Mn₃Sn while the latter is half the value. As shown in the current dependence of Hall resistance displayed in Fig. 3b-c and 3e-f, deterministic current-induced switching is also observed in both structures with a switching ratio around 11% and 20%, respectively. It is interesting to note that the switching polarity of Ti/Mn₃Sn is the same as that of Ta/Mn₃Sn but opposite of Ta/MgO/Mn₃Sn with a thick MgO spacer and MgO/Mn₃Sn.”

e) Page 10, line 226-230:

Added: “Besides, we also performed current-induced switching in Mn₃Sn directly deposited on Si/SiO₂ substrate (Fig. 3h and 3i). The much smaller remanence in its AHE loop (Fig. 3g) as compared to other structures above indicate that grain size and its orientation in Si/SiO₂/Mn₃Sn can be quite different compared to other samples, resulting in negligible current-induced switching.”

f) Moved Fig. 2g-i in original manuscript to Fig. 3. Added figures for AHE and current-induced switching curves of MgO/Mn₃Sn and Si/SiO₂/Mn₃Sn in Fig. 3.

Comment 2: “The inter-grain spin-transfer torque requires the spin current to flow across different grains and the magnetic switching behavior only occur in certain grains, which is also shown in the MOKE experiments in Fig. 3. During the sputtering process, the sizes and distributions of the grains are unable to be uniform. It will cause inhomogeneity when applying noncollinear AFM to high-density devices which is the main driving force for the investigation of noncollinear AFM. How to solve the inhomogeneity problem?”

Response: Thank the reviewer for the very insightful observations and comment. Indeed, inhomogeneity can be a challenging issue for high-density memory applications. As revealed in the cross-sectional TEM images of Ta/Mn₃Sn(12)/Ta (Fig. 2a) and Ta/MgO/Mn₃Sn(12)/MgO/Ta (Fig. 2b), the grain size of Mn₃Sn is in the range of 10 nm to 40 nm, which is undoubtedly not suitable for conventional bitwise memory applications. However, when the grain size and distribution are optimized, such kind of structure may also be advantageous for applications which require the output to be a non-linear function of the input, *e.g.*, sigmoid function. This is possible because grains with different size and orientation will be switched at different driving current and the switching will gradually slow down and eventually stop at a sufficiently large current. Take the Hall bar used in this study as an example (Hall cross: 15 μm \times 5 μm), assuming an average grain size of 30 nm, the number of grains in the Hall bar cross area will be around 8.3×10^4 . Such a large number of grains allows us to obtain consistent anomalous Hall effect and current-induced switching results in repeated samples of same structure, and the switching ratio indeed shows a nonlinear dependence on the driving current. It is worth pointing out that, after we have submitted the manuscript, non-uniform switching of Mn₃Sn grains has been revealed directly using NV magnetometry in a recent study on current-induced switching in heavy metal/Mn₃Sn bilayers, as shown in Fig. 2c (G. Q. Yan, et. al., Adv. Mater. 2200327 (2022)). The results are consistent with our observations.

FIG. 2. a-b, Cross-sectional TEM images of Ta(2)/Mn₃Sn(12)/Ta(3) and Ta(2)/MgO(4)/Mn₃Sn(12)/MgO(4)/Ta(1.5), respectively. **c,** Image of the SOT-induced change of the out-of-plane stray field ΔB_z from Mn₃Sn/Pt (figure taken from (G. Q. Yan, et. al., Adv. Mater. 2200327 (2022))).

Coming back to the reviewer’s question as how to solve the inhomogeneity problem (required for particular applications), we do have some rough ideas which might be explored in future studies. One is to intentionally introduce non-magnetic materials to separate the grains, as what has been done in producing high-density and low noise magnetic recording media in the past several decades. When properly chosen and optimized, the grain boundary segregation material may facilitate the control of both grain size and its distribution, as well as the coupling between neighbouring grains. Some of the methods developed for bit patterned media fabrication may also be explored and tailored for producing Mn₃Sn with uniform grain size. The second possible approach is to form patterned seed layer followed by the deposition of Mn₃Sn. In this case, grain boundary may be pinned at the edge of the patterned seed layer. If one is only concerned about the position of the gain boundary rather than the grain size, one may also try substrates with well-defined terraces or grain boundaries (*e.g.*, the use of bi-crystal substrate).

Having said that we would like to point out that the focus of this study is to reveal an alternate mechanism for current-induced switching in polycrystalline Mn₃Sn, which itself is important for any intended applications, bit memory or other types of devices. Non-collinear AFM based tunnel junctions have been proposed in several theoretical papers (J. Železný, et. al., Phys. Rev. Lett. 119, 187204 (2017); D. Go et al., arXiv:2201.11476; J. Dong et al., arXiv:2112.06568), arguing that the spin chirality of the constituent AFM can be switched by spin-polarized or chiral spin current. However, so far, such kind of device has not been demonstrated experimentally. Our work provides direct evidence that the proposed tunnel junction is feasible, and particularly it would be more

interesting if the crystalline orientation of the two electrodes can also be controlled using different types of seed layers.

In response to the reviewer's comment, we have made the following changes:

g) Page 21, line 438-445:

Added: "The random distribution of crystal grain orientation and large size distribution may pose a challenge in applications where the device size has to be very small, *e.g.*, non-volatile memory. In this regard, some of the techniques that have been developed for high-density magnetic recording media may be used to reduce the grain size and improve its size distribution. In addition, templated seed layer or grain boundary materials may be employed to control the crystalline orientations. On the other hand, nonuniformity in grain size distribution may also be exploited for emerging applications where gradual instead of abrupt switching is required, such as neuromorphic computing."

h) Added TEM images in Fig. 6.

Comment 3: "How to control the switching polarity in the polycrystalline situation? For example, the number of grains which generate +y spin polarization and that of grains generating -y spin polarization may be equal, or the former is larger than the latter sometimes while is smaller in other times."

Response: Thank the reviewer for raising a very important point. As described in the main text, we use the polarizer/analyzer model to explain the current-induced switching via the inter-grain spin-torque transfer mechanism. To achieve current-induced switching which is detectable by AHE, the configuration for the polarizer and analyzer must meet the following requirements. Firstly, the direction of spin polarization needs to be parallel with the kagome plane of analyzer as perpendicular-to-kagome-plane polarization will only induce chiral-spin rotation (H. Tsai et al, Nature 580, 608 (2020); Y. Takeuchi, et. al., Nat. Mater. 20, 1364 (2021)). Secondly, based on the AHE measurement geometry revealed in experiments using single crystals (S. Nakatsuji et al, Nature 527, 212-215, (2015)), large AHE signals are obtained when $B \parallel [01\bar{1}0], I \parallel [0001]$ or $B \parallel [2\bar{1}\bar{1}0], I \parallel [01\bar{1}0]$, and the Hall voltage is measured in the direction perpendicular to both the magnetic field and current direction. In both configurations, the magnetic field is in the kagome plane because it is easy to switch the polarization of spin sublattices or the so-called octupole moment. In the inter-grain spin-torque transfer model presented in this study, the polarization of spin current from the polarizer is equivalent to the B -field, as shown in Eq. (1) of the main text. The spin current entering a particular Mn_3Sn grain is the sum of spin current or spin polarized current from all neighbouring grains. As the

thickness of Mn_3Sn used in the switching experiments is much smaller than the average grain size, we can safely assume that there is dominantly only a single grain in the thickness direction. In other words, we only need to consider spin current from grains on the same film plane. In this case, the average effects of spin current perpendicular to the charge current should be cancelled out based on symmetry considerations. Therefore, we only need to consider the longitudinal spin current, or spin polarized current along the charge current direction.

For a collinear ferromagnet, the equilibrium spin current polarization is generally accepted to be parallel with the local magnetization direction. However, recent studies suggest that transverse polarization component can also exist (*e.g.*, A. Davidson et al., Phys. Lett. A 384, 12628 (2020)). The polarization of spin current in noncolinear AFM like Mn_3Sn is more complicated. Theoretical studies showed that transverse (like spin Hall current), longitudinal (akin to spin polarized current), and chiral spin current can exist in non-collinear AFM (Y. Zhang, et. al., Phys. Rev. B 95, 075128 (2017); J. Železný, et. al., Phys. Rev. Lett. 119, 187204 (2017); D. Go et al., arXiv:2201.11476). For the present case, we believe the longitudinal or spin polarized current plays the dominant role. Now, the question is what is the polarization direction of spin polarized current? According to J. Železný, et. al., with the presence of spin-orbit interaction, there is always an in-kagome-plane polarization for the spin polarized current. If we ignore onsite chirality, the average spin polarization may be represented by the net magnetization direction.

As revealed by XRD and magnetic measurements, the crystalline grains may be divided into three groups based on the angle between the hexagonal plane and the substrate: i) $(11\bar{2}0)$ and $(2\bar{0}20)$, ii) $(10\bar{1}1)$, $(20\bar{2}2)$ and $(20\bar{2}1)$, and iii) (0002) . Grains with (0002) orientation do not contribute to the measured AHE signal and therefore, it cannot be an analyzer. The configurations in Fig. 3a and 3b are the two configurations discussed in the original manuscript, where both polarizer and analyzer are group i) grains. In general, when the charge current passes through the analyzer, spin current with both transverse and longitudinal polarization is generated. In the case of configurations shown in Fig. 3a and 3b, the longitudinal spin polarization does not cause magnetization switching as it is perpendicular to the kagome plane; therefore, we only indicated the spin current with transverse spin polarization in the original manuscript. However, when both group ii) and iii) grains are considered, there are more polarizer/analyzer configurations which can result in switching of the analyzer with detectable AHE. To be more inclusive, we include spin current with both transverse and longitudinal polarization (or spin polarized current) in Fig. 3a-e below (corresponding to Fig. 5b-f in the revised manuscript) and denote it as j_s . Under this notion, j_s in Fig. 3a and Fig. 3b may be written as $j_s = j_{x'x'}^{x'} + j_{x'x'}^{y'}$ and $j_s = j_{z'z'}^{x'} + j_{z'z'}^{y'}$, respectively. As discussed in the original manuscript, $j_{x'x'}^{y'}$ and $j_{z'z'}^{y'}$ are the spin current with transverse spin polarization which

will cause switching of the magnetization, whereas $j_{x'x'}$ and $j_{z'z'}$ are spin current with longitudinal polarization which would only induce spin rotation in the kagome plane. For configurations shown in Fig. 3c-3e, however, it is difficult to decompose j_s into spin current with longitudinal and transverse polarizations in the local coordinates as the kagome planes of the polarizer and analyzer do not form a right-angle. In these cases, depending on the exact kagome plane orientation of the polarizer and analyzer, spin currents with both transverse and longitudinal spin polarization may cause the switching of the magnetization. In the current measurement configuration, phenomenologically the AHE signal is proportional to the vertical component of the net magnetization in the kagome plane. Therefore, the configurations shown in Fig. 3c-e would lead to switching with a polarity opposite to that of Fig. 3a and 3b, including other possible configurations.

Fig. 3. a-e Configurations of crystalline planes for a pair of polarizer and analyzer in inter-grain STT-based switching. $x'y'z'$ are local coordinate for polarizer. x', y' are in the kagome plane, and z' is perpendicular to the kagome plane. H_x (H_y) is the assistive along x -axis (y -axis) of the global coordinate, whereas $H_{x'}$ is the projection of H_x in the local coordinates for polarizer.

As we discussed in both the main text and supplementary information of the original manuscript, the texture of polycrystalline Mn_3Sn is well reflected in its hysteresis loop measured along z -axis. As shown in Fig. 1d of the original manuscript, the M - H loops can be decomposed into contributions from two groups of grains. In the case of Ta/MgO(t_{MgO})/ Mn_3Sn , the amount of group i) grains

increases with increasing the MgO thickness. The group i) to ii) grains ratio increases from 21% at $t_{\text{MgO}} = 0$, to 61% at $t_{\text{MgO}} = 2$ nm, and 83% at $t_{\text{MgO}} = 4$ nm. As shown in Fig. 4 below, the M - H loop for Ti(2)/Mn₃Sn(60) is very similar to that of Ta/MgO(2)/Mn₃Sn, at which sign reversal of switching has not occurred yet (Fig. 2e of original manuscript), *i.e.*, the switching polarity is the same for Ta/Mn₃Sn, Ti/Mn₃Sn, and Ta/MgO/Mn₃Sn with small MgO thickness.

Fig. 4. a, Out-of-plane M - H curve with the fitting for Ti(2)/Mn₃Sn(60). **b**, Decomposed sub-loops of the $M - H$ loop in **a**.

Fig. 5. **a-c**, In-plane $M - H$ loops measured at different field angle φ_H with respect to the long axis of a rectangular sample for MgO(2)/Mn₃Sn(60), MgO(4)/Mn₃Sn(60) and Ti(2)/Mn₃Sn(60), respectively. **d**, Decomposed sub-loops of the $M - H$ loops in **a-c**.

In order to gain further insights into the grain-based switching mechanism, we measured the in-plane $M-H$ loops of MgO(2)/Mn₃Sn(60), MgO(4)/Mn₃Sn(60), and Ti(2)/Mn₃Sn(60) at selected field angles φ_H and the results are shown in Fig. 5a-c. The samples are cut into a rectangular shape, and we use the long axis of the sample as a reference, *i.e.*, $\varphi_H = 0$ corresponds to the case that the applied field is along the long axis of the sample. As the substrate was rotated continuously during the deposition, the grain distribution on the film plane is presumably isotropic and, therefore, the use of long axis as a reference is just for convenience; it does not make any difference if the short axis or any other direction is used to indicate the field angle. As expected, the film does not show any evident in-plane

anisotropy in terms of grain distributions. The slight differences in the M - H loops at different angles might be caused by the finite out-of-plane field component due to sample misalignment.

We also performed XRD measurements for Ti/Mn₃Sn(12) and MgO/Mn₃Sn(12), where the Mn₃Sn thickness is same as the one used for current-induced switching. As can be seen from Fig. 6a-b, MgO/Mn₃Sn(12) shows obvious peaks of (11 $\bar{2}$ 0), which is missing in Ti/Mn₃Sn(12), while the peak of (20 $\bar{2}$ 1) exists in both samples. In addition, there is also a very small peak of (20 $\bar{2}$ 0) in both samples and an another small peak of (10 $\bar{1}$ 1) in Ti/Mn₃Sn(12). This indicates that there are more group i) grains with kagome plane perpendicular to the film plane in MgO/Mn₃Sn(12), which is consistent with the out-of-plane M - H measurement results.

Fig. 6. a-b, XRD profiles of Ti/Mn₃Sn(12) and MgO/Mn₃Sn(12), respectively.

In addition, we also examined the temperature dependence of longitudinal resistivity and Hall conductivity of 12 nm Mn₃Sn with different seed layers (MgO and Ti underlayers with different thickness t_{Ti} and t_{MgO}) to gain some insight of the seed layer effect on its crystalline orientations. Figure 7a and 7b show the relative change of longitudinal resistivity $\rho_{xx}/\rho_{xx,300K}$ with respect to the value at 300 K and Hall conductivity $\sigma_H(H = 0)$ at zero field (corresponding to AHE contributed by non-collinear AFM phase) as a function of temperature, respectively. As can be seen, overall, except Ti(5)/Mn₃Sn, all the samples show decreased ρ_{xx} at lower temperature. As for Hall

conductivity, phase transition of non-collinear is observed for all the structures. However, Mn_3Sn grown on Ti layer shows higher transition temperature of around 260 K compared to $\text{MgO}/\text{Mn}_3\text{Sn}$ with transition temperature around 220 K. According to temperature dependence of σ_H for single crystal (field applied along kagome plane) and polycrystalline Mn_3Sn , single crystal has much lower transition temperature at around 50 K compared to the latter (S. Nakatsuji, et. al., Nature 527, 212 (2015); T. Higo, et. al., Appl. Phys. Lett. 113, 202402 (2018)). This implies that the lower transition temperature in $\text{MgO}/\text{Mn}_3\text{Sn}$ might be due to the more group i) grains, whose kagome planes are parallel to z-axis. This corroborates well with the M - H and XRD measurement results shown above. From the temperature dependent AHE, we also find that $\text{MgO}/\text{Mn}_3\text{Sn}$ still shows negative AHE at low temperature (50 K to 108 K) as shown in Fig. 7d, while $\text{Ti}/\text{Mn}_3\text{Sn}$ shows ferromagnetic type of AHE (Fig. 7e-f), suggesting the Mn_3Sn transition into glassy FM phase (J. M. Taylor, et. al., Phys. Rev. B 101, 094404 (2020)). As the increase of Ti thickness, the amplitude of FM-phase AHE at low temperature further increases. In addition, as shown in Fig. 7g-i, $\text{Ti}/\text{Mn}_3\text{Sn}$ also shows evident magnetoresistance (MR) at 50 K, which is however absent in $\text{MgO}/\text{Mn}_3\text{Sn}$, further indicating the FM phase in $\text{Ti}/\text{Mn}_3\text{Sn}$ at low temperature. This magnetic property of $\text{Ti}/\text{Mn}_3\text{Sn}$ corresponds well with that of epitaxial Mn_3Sn (0001) film (J. M. Taylor, et. al., Phys. Rev. B 101, 094404 (2020)), though $\text{Ti}/\text{Mn}_3\text{Sn}$ still shows negative AHE at room temperature while Mn_3Sn (0001) film only shows ordinary Hall effect at room temperature. This indicates that the Ti seed layer facilitates the formation of (0001) crystalline plane (with c -axis along z -direction) in Mn_3Sn , especially a thick Ti layer. For $\text{Ti}/\text{Mn}_3\text{Sn}$ with thinner Ti layer ($t_{\text{Ti}} = 1, 2$ nm), the relatively large AHE at room temperature can be explained by the fact that the crystalline planes of Mn_3Sn are mostly $(10\bar{1}1)$, $(20\bar{2}2)$ and $(20\bar{2}1)$ (group ii), whose c -axis is tilted towards but not completely aligned to z -direction. This again suggests that the different seed layers lead to different crystalline orientations in Mn_3Sn , *e.g.*, Mn_3Sn with MgO seed layer has more group i) grains with $(11\bar{2}0)$ and $(20\bar{2}0)$ orientations, and Mn_3Sn with Ti seed layer has more group ii) and group iii) grains with $(10\bar{1}1)$, $(20\bar{2}2)$, $(20\bar{2}1)$, and (0002) orientations.

At last, depending on the exact polarization direction of the spin current from a particular polarizer, the magnetization direction of its neighbouring grain in the current direction may be switched into one of its three equivalent easy axis direction in the kagome plane. The switching may begin with one of the spin-sublattices first, followed by the other two (depending on the angle between sublattice spin and spin current polarizations). Once the polarization of analyzer is switched, it may now function as a polarizer to switch the next grain (depending on their mutual spin configurations), leading to switching of large cluster of grains. At steady state, there will be a

distribution of net magnetization direction among all the grains. The AHE voltage measured will be proportional to the average of projection of the magnetization of each grain along z -direction.

Therefore, to sum up, the switching polarity difference in MgO/Mn₃Sn and Ti/Mn₃Sn may be due to the different crystalline orientations in Mn₃Sn affected by seed layer. In MgO/Mn₃Sn, there are more group i) grains and therefore the switching configuration in Fig. 3a-b dominates, while for Ti/Mn₃Sn the switching configuration in Fig. 3c-e becomes dominant due to more group ii) and group iii) grains. A quantitative discussion, however, is out of the scope of this work.

Fig. 7. **a**, Relative change of longitudinal resistivity $\rho_{xx}/\rho_{xx,300K}$ with respect to the value at 300 K as a function of temperature. **b-c**, Hall conductivity at zero field σ_H ($H = 0$) of Mn₃Sn with different seed layers as a function of temperature. **d-f**, Hall resistivity as a function of magnetic field for MgO(2)/Mn₃Sn(12), Ti(2)/Mn₃Sn(12), and Ti(5)/Mn₃Sn(12), respectively. **g-i**, Longitudinal resistance as a function of field at 50 K for MgO(2)/Mn₃Sn(12), Ti(2)/Mn₃Sn(12), and Ti(5)/Mn₃Sn(12), respectively.

In response to the reviewer's comment, we have made the following changes:

i) Page 16-18, line 336-390:

Original: “Therefore, for simplicity, we only need to consider the contributions from two side-grains transverse to the current and one along the current direction. The two side-grains inject transverse spin current, whereas the other grain injects spin polarized current into the analyzer. As discussed in detail in the Supplementary Information S6, among the twenty-four possible configurations, two of them will lead to spin rotation in the kagome plane, whereas the other two will lead to magnetization switching, which can be detected by the Hall measurement. Figure 4b and c show the two possible configurations for deterministic switching. Here, the primed coordinate system is the local coordinate system attached to the polarizer such that x' and y' are always in the kagome plane, which is to be differentiated from the global coordinate system shown in Fig. 4a. In the configuration shown in Fig. 4b, an assistant field H_x is used to align \mathbf{m} of the polarizer along x -direction. By doing so, a spin polarized current with polarization along the y' - direction will be induced by a charge current in x -direction (note x and x' are in the same direction in this case). By convention, we use $j_{\beta\gamma}^\alpha$ to denote the spin current flowing in β direction with a polarization along α , induced by a charge current in γ direction. When the spin-polarized current enters the analyzer, it will lead to magnetization switching due to the STT. The switching polarity should depend on the delicate balance of the spin-polarized current from neighbouring grains. This may explain the seed layer dependence of switching polarity observed experimentally. On the other hand, in the configuration shown in Fig. 4c, the polarizer and analyzer inter-change their roles, in which STT-based switching is also possible when an assistant field is applied in y -direction. However, as the spin polarization in the two cases may be different in amplitude, the ratio of switched regions can differ significantly in the two cases. In actual samples, the polarizer and analyzer are not necessarily always perpendicular to each other; in fact, they can form any angle as inferred from the in-plane magnetic measurements. In addition, some of the kagome planes are tilted from z -axis, and therefore, the assistant field angle dependence should not follow exactly that of case (i).”

Revised: “Furthermore, the average effects of spin current transverse to the charge current should be largely cancelled out based on symmetry considerations. Therefore, only longitudinal spin current or spin-polarized current is considered to possibly contribute to switching. For a collinear ferromagnet, the equilibrium polarization of spin-polarized current is generally parallel with the local magnetization direction. However, recent studies suggest that transverse polarization also exists. For the present case, the longitudinal or spin-polarized current plays the dominant role. According to J. Železný, et. al., with the presence of spin-orbit interaction, the spin polarized current emanating from the polarizer contains both a transverse and a longitudinal polarization. As revealed by XRD and magnetic measurements, the crystalline grains in polycrystalline Mn_3Sn may be divided into three

groups based on their kagome plane orientation with respect to the substrate: i) $(11\bar{2}0)$ and $(20\bar{2}0)$, ii) $(10\bar{1}1)$, $(20\bar{2}2)$ and $(20\bar{2}1)$, and iii) (0002) . Grains with (0002) orientation do not contribute to the measured AHE signal and therefore, it cannot be analyzer. In general, when the charge current passes through the analyzer, spin current with both transverse and longitudinal polarization is generated. In the case of configurations shown in Fig. 5b,c, only spin current with transverse polarization plays a role in switching, because the longitudinal spin polarization is perpendicular to the kagome plane. Here, the primed coordinate system is the local coordinate system attached to the polarizer such that x' and y' are always in the kagome plane, which is to be differentiated from the global coordinate system shown in Fig. 5a. In the configuration shown in Fig. 5b, an assistant field H_x is used to align \mathbf{m} of the polarizer along x -direction. In this case the spin current from the polarizer may be written as $j_s = j_{x'x'}^{x'} + j_{x'x'}^{y'}$ (note x and x' are in the same direction in this case). By convention, we use $j_{\beta\gamma}^\alpha$ to denote the spin current flowing in β direction with a polarization along α , induced by a charge current in γ direction. When the spin-polarized current enters the analyzer, spins with transverse polarization $j_{x'x'}^{y'}$, will lead to magnetization switching due to the STT. The switching polarity should depend on the delicate balance of the spin-polarized current from neighbouring grains. On the other hand, in the configuration shown in Fig. 5c, the polarizer and analyzer inter-change their roles, in which STT-based switching is also possible when an assistant field is applied in y -direction (See Supplementary Information S9). However, as the spin polarization in the two cases may be different in amplitude, the ratio of switched regions can differ significantly in the two cases. In actual samples, the polarizer and analyzer are not necessarily always perpendicular to each other as revealed by both XRD and SQUID results. Figure 5d-f show three possible configurations which can lead to both magnetization switching and detectable AHE. Unlike cases in Fig. 5b and 5c, it is difficult to decompose j_s into spin current with longitudinal and transverse polarizations in the local coordinates as the kagome planes of the polarizer and analyzer do not form a right-angle. In these cases, depending on the exact kagome plane orientation of the polarizer and analyzer, spin currents with both transverse and longitudinal spin polarization may cause the switching of the magnetization. In the current measurement configuration, phenomenologically the AHE signal is proportional to the vertical component of the net magnetization in the kagome plane. Therefore, the configurations shown in Fig. 5d-f would lead to switching with a polarity opposite to that of Fig. 5b and 5c.

As discussed earlier, the texture of polycrystalline Mn_3Sn is well reflected in its M - H loop. From the decomposition of M - H curves as shown in Fig. 1d, we find that in the case of $\text{Ta/MgO}(t_{\text{MgO}})/\text{Mn}_3\text{Sn}$, the contribution from group i) grains increases with increasing the MgO

thickness, indicating that MgO seed layer facilitates the formation of group i) grains and the switching in MgO/Mn₃Sn might be dominated by the configuration in Fig. 5b and 5c. In addition, as shown in Supplementary Information S2, the *M-H* loop for Ti(2)/Mn₃Sn(60) is very similar to that of Ta/MgO(2)/Mn₃Sn, at which sign reversal has not occurred yet, *i.e.*, the switching polarity is the same for Ta/Mn₃Sn, Ti/Mn₃Sn, and Ta/MgO(2)/Mn₃Sn. The XRD profiles and temperature dependent AHE of Mn₃Sn in different structures further suggest that there are mostly group ii) and iii) grains in Ti/Mn₃Sn (Supplementary Information S1 and S7). Therefore, the switching configurations in Fig. 5d-f may become dominant in Ti/Mn₃Sn. At this stage, however, a quantitative analysis of the switching polarity would be difficult considering the random distribution of the crystalline orientation of the grains.”

- j) Revised the schematics of SOT and IGSTT based switching in Fig. 5 and the caption.
- k) Added XRD profiles for Ti/Mn₃Sn(12) and MgO/Mn₃Sn(12) in Supplementary Information S1.
- l) Added *M-H* curve of Ti/Mn₃Sn in Supplementary Information S2.
- m) Added temperature dependence of longitudinal resistivities and Hall conductivities for different structures in Supplementary Information S7.
- n) Removed Supplementary Information S6 from the original Supplementary.

Comment 4: “The assistant field is comparatively large. Is there any critical assistant field for the present Mn₃Sn. In general, the assistant field is used to break the symmetry from the macrospin viewpoint or conquer the DMI field from the multi-domain viewpoint. What is the function of the assistant field here?”

Response: We agree with the observation of the reviewer that the assistant field indeed is relatively large. Figure 8a shows the current-induced switching loops at varying $\mu_0 H_x$ from -100 mT to 100 mT, with the extracted switching ratio displayed in Fig. 8b. As can be seen, there is almost no switching occurred at zero assistant field. As the field increases, the switching amplitude gradually increases and reaches a peak at 60 mT, after which it slowly decreases with further increase of the assistant field. The field-dependence of the current-induced switching is consistent with those reported for heavy metal/Mn₃Sn bilayer structures (H. Tsai, et. al., Nature, 580, 608 (2020); G. Q. Yan, et. al., Adv. Mater. 2200327 (2022); G. K. Krishnaswamy, et. al., arXiv:2205.05309). Clearly, there is no critical assistant field for switching.

Indeed, for conventional HM/FM bilayers with PMA, there is a need to use an in-plane assistant field to break the symmetry (H. Tsai, et. al., Nature, 580, 608 (2020); S. Ghosh, et. al., Phys. Rev. Lett. 128, 097702 (2022)). However, in the present case, we believe the main role of the

assistant field is to change the spin polarization direction of the polarizer rather than breaking the symmetry of the analyzer. This is because the spin polarization is already in the kagome plane of the analyzer, and one of the sub-lattices will be switched first, after which the other two will follow. This is similar to the HM/FM bilayer case where there is a z -component of polarization in the spin current, which allows for field-free switching. Apart from the damping-like torque, as shown in Eq. (1) of the main text, filed-like torque may also help to switch the spin sublattices without resorting to any assistant field. The gradual increase of switching ratio with the assistant field is consistent with the scenario that the degree of alignment of magnetization to external field direction depends on the orientation of the individual grains. The relatively large assistant field for maximum switching ratio is expected from the large coercivity in M - H loops, especially for tilted grains. The decrease of switching ratio at large field suggests that too large an assistant field will impede the rotation of spin sublattices of the analyzer. Therefore, although the role of assistant field is different in conventional SOT and IG-STT based switching, both are required for controlling the switching polarity. At the end of the day, whether the SHE and self-induced spin current combined will enhance or reduce the switching ratio depend on their respectively switching polarity.

The DMI is known to be non-negligible in Mn_3Sn , but it is mostly for defining and stabilizing the inverse triangular spin configuration (T. Nagamiya, et. al., *Solid State Commun.* 42, 385 (1982); Y. Yamane, et. al., *Phys. Rev. B* 100, 054415 (2019); H. C. Zhao, et. al., *Nat. Commun.* 12, 5266 (2021)). In addition, from the MOKE images shown in Supplementary Fig. 8 as well as the nitrogen-vacancy magnetometry images (Fig. 8c) reported in the recent study ((G. Q. Yan, et. al., *Adv. Mater.* 2200327 (2022)), we can see that the switch of magnetic state in Mn_3Sn with increasing the current is grain-based without domain wall nucleation/propagation. Therefore, the involvement of domain wall, if any, is minimum.

FIG. 8. a-b, Current dependence of Hall resistance and extracted switching ratio $\Delta R_H/R_{AHE}$ of Ta(2)/MgO(3)/Mn₃Sn(12) at varying $\mu_0 H_x$ from -100 mT to 100 mT. **c,** Imaging of SOT-induced variation of the stray field ΔB_z in Mn₃Sn/Pt at different states as the increase of injected current amplitude (figures taken from G. Q. Yan, et. al., Adv. Mater. 2200327 (2022)).

In response to the reviewer’s comment, we have made the following changes:

- o) Added current-induced switching loops at different in-plane assistant field in Supplementary Information S6.

Comment 5: “Whether the switching ratio is seriously reduced by polycrystalline feature with variable kagome planes? Whether fully switching (when the thermal effect is not taken into account) can be realized for the SOT switching of Mn₃Sn?”

Response: As switching only occurs when the spin polarization is within the kagome plane, it is not possible to achieve 100% switching in polycrystalline samples as revealed by previous studies including the recent study using the nitrogen-vacancy sensor (G. Q. Yan, et. al., Adv. Mater. 2200327 (2022)). For HM/Mn₃Sn bilayer, where the spin current is mainly from heavy metal layer and the spin polarization direction is fixed, the more kagome planes that is parallel to the spin polarization the larger the switching ratio will be, but it can’t reach 100%. For simplicity, we assume that the

sample consists of group i) grains with randomly distributed orientations. In this case, we may estimate the switching ratio of HM/Mn₃Sn bilayers by assuming 1) the switching is based on conventional damping-like SOT, 2) only grains with crystalline plane coplanar with the spin polarization can be switched, and 3) grains which meet the polarization requirement are completely switched at sufficiently large current, which turned out to be $\frac{2}{\pi} \int_0^{\pi/2} \cos \varphi d\varphi = 0.64$. Here, φ is the misalignment angle of *c*-axis of kagome plane and spin polarization from HM metal layer. The actual switching ratio can be smaller than the estimated value when grains with tilted kagome planes, *e.g.*, group ii) grains, are considered equally. The estimated switching ratio is in good agreement with the experimental results reported by other groups (H. Tsai et al, Nature 580, 608 (2020); H. Tsai, et. al., AIP Adv. 11, 045110 (2021); H. Tsai, et al., Small Sci. 1, 2000025 (2021); G. Q. Yan, et. al., Adv. Mater. 2200327 (2022)). In fact, the maximum switching ratio for HM/Mn₃Sn bilayer structure is around 50% in previous study (H. Tsai, et. al., Small Sci. 1, 2000025 (2021)) and 61% in the present work, which is close to the estimated value. The smaller switching ratio in previous studies might be due to the larger Mn₃Sn thickness used (30-40 nm).

On the other hand, when the switching is mainly induced by the self-generated spin current in Mn₃Sn, the switching ratio can be much smaller because a pair of neighbouring grains functioning as polarizer and analyzer with specific crystalline orientation is required. From current-induced switching in different structures, we find that the switching ratio overall presents a positive correlation with the AHE remanence $R_{H(H=0)}/R_{AHE}$ as shown in Fig. 9, which is an indicator of degree of texturing of the non-collinear AFM film. The largest switching ratio achieved is around 20% in Ti/Mn₃Sn, where the AHE remanence is also largest. In addition, when the IG-STT and conventional SOT mechanism co-exist, the combined switching ratio of HM/Mn₃Sn may become larger or smaller depending on the polarity of the switching induced by the two mechanisms. Nevertheless, we believe 100% switching is possible for small samples with a single grain in Ta/Mn₃Sn bilayer (via spin Hall current) or 2 grains in Mn₃Sn (via inter-grain spin torque transfer).

FIG. 9. The remanence $R_{H(H=0)}/R_{AHE}$ dependence of the switching ratio for Mn₃Sn with different underlayers. The thickness of Mn₃Sn fixed as 12 nm for all the structures.

In response to the comment, we have made the following change:

p) Page 21, line 445-451:

Added: “One other limitation is the incomplete switching, which is also reported in the other studies on HM/Mn₃Sn systems. From the current-induced switching in different structures, we find that the switching ratio overall presents a positive correlation with the AHE remanence $R_{H(H=0)}/R_{AHE}$ (as shown in Supplementary Information S10) which is an indicator of degree of texturing of the non-collinear AFM film. Therefore, further studies may be required to produce well textured films with desirable grain orientation for current-induced switching.”

q) Added Supplementary Information S10.

Comment 6: “The SOT switching was realized in ferromagnetic single layer (Nature Nanotechnol. 14, 819 (2019), Phys. Rev. B 101, 214418 (2020), Adv. Funct. Mater. 2020, 2005201) and antiferromagnetic single layer (Science 351, 587 (2016), Nat. Commun. 9, 348 (2018), Nat. Mater. 18, 931 (2019)). This point should be mentioned in this work. And the origin for the SOT effect in different magnetic single layer should be compared.”

Response: We didn’t cite these papers because all these materials have colinear spin configuration. Following the reviewer’s suggestion, we have added these references with brief comment on their origin in the revised manuscript.

In response to the reviewer’s comment, we have made the following change:

r) Page 21, line 452-460:

Added: “Before we conclude, it is worth mentioning that the proposed mechanism is different from previous reports on magnetization switching of single layer FM or AFM which treat the single layer largely homogeneous in terms of the magnetic order parameter. The mechanisms proposed in these early works include spin current with spin polarization transverse to the magnetization AHE-induced surface spins with spin polarization rotation in FM, asymmetric absorption of internally generated spin Hall current in ferrimagnet, and current-induced internal staggered spin-orbit torque due to local inversion asymmetry of spin sublattices in colinear AFM. However, all these mechanisms cannot account for the results obtained in this study.”

C) Response to comments from Reviewer #2

Comment 1: “The switching polarity is dependent on the normal metal (NM) layer, i.e., whether it is MgO or Ti, attached to Mn₃Sn. This may be indicating that the NM/Mn₃Sn interface is playing the key role, instead of the self-generated torque in Mn₃Sn. For example, whereas the systems in the present work may not be exactly known for exhibiting a large Rashba spin-orbit coupling, it is symmetrically allowed to appear here and it would also lead to the magnetization switching via the corresponding spin-orbit torques. The authors are not discussing this possibility. Some more systematic study is required to exclude possible influences of the interface.”

Response: We thank the reviewer for mentioning the possible role of interface. Firstly, in the MgO thickness dependent study of switching in Ta/MgO/Mn₃Sn/MgO/Ta, we always kept the bottom and upper MgO layer the same thickness to minimize the induced structure asymmetry. If there is any significant Rashba effect between MgO and Mn₃Sn interface, the effects of the upper and lower interface would be largely cancelled out and therefore wouldn't lead to switching with such noticeable amplitude as shown in our results. In addition, we have also fabricated samples with a structure of Si/SiO₂(substrate)/Mn₃Sn/MgO/Ta. However, as shown in Fig. 1b-c of the rebuttal, there is almost no current-induced switching occurred. On the contrary, when a MgO layer is added underneath the Mn₃Sn, evident switching with a ratio of around 11% is observed (Fig. 1e-f of the rebuttal). Compared with Si/SiO₂/MgO/Mn₃Sn/MgO/Ta structure, Si/SiO₂/Mn₃Sn/MgO/Ta has a higher structural asymmetry and thus should have a larger switching ratio if Rashba effect is the dominant switching mechanism. Instead, it shows negligible switching after current injection. Therefore, it is unlikely that the Rashba effect plays a role here, at least not the dominant role. The other evidence is that, similar as the SHE-induced spin current, the spin accumulation or spin current from Rashba effect is also global in Mn₃Sn, and should lead to as uniform switching in

Ta/MgO(4)/Mn₃Sn sample as the Ta/MgO(2)/Mn₃Sn case. However, we can see from Fig. 11a and Fig. 11b that the switching in Ta/MgO(4)/Mn₃Sn is much less uniform than the switching in Ta/MgO(2)/Mn₃Sn sample mainly contributed by SHE-induced spin current.

In addition, as has been mentioned by the reviewer, sizable Rashba spin-orbit coupling at Mn₃Sn interfaces has not been reported. In a recent study (H. Tsai, et. al., AIP Adv. 11, 045110 (2021)), the researchers inserted a Cu layer with large diffusion length between Mn₃Sn and heavy metal to remove the heavy metal/Mn₃Sn interface. They found that the switching in heavy metal/Cu/Mn₃Sn shows same switching polarity and ratio as heavy metal/Mn₃Sn case, which excluded the Rashba effect in current-induced switching of Mn₃Sn. In addition, the charge-to-spin conversion in Ti/FM has been reported to be negligible from both SOT characterization and current-induced switching measurement (L. Zhu, et. al., Phys. Rev. Appl. 15, L031001 (2021); Y. Yang, Phys. Rev. Appl. 13, 034072 (2020)), so it is also unlikely that there is significant spin-current generation or spin accumulation at Ti/Mn₃Sn interface. Nevertheless, in response to the reviewer's comment, to exclude the interface effect at Ti/Mn₃Sn, we also added samples of Ti(t_{Ti})/Mn₃Sn(12) with different Ti thickness t_{Ti} from 1 nm to 5 nm, and Ti(1)/Mn₃Sn(12)/Ti(1) with symmetric structure. Figure 10a shows the AHE resistivity of Ti(t_{Ti})/Mn₃Sn(12), from which the AHE of Mn₃Sn keeps decreasing as the increase of Ti thickness. When t_{Ti} is larger than 2 nm, the AHE becomes negligibly small and almost submerged by ordinary Hall effect signal (as shown in Fig. 10b), which might be because the Ti layer facilitates the formation of (0001) crystalline plane (with c -axis along z -direction) in Mn₃Sn, especially a thick Ti layer, as has been shown in Fig. 7 of the rebuttal. Therefore, we could only perform current-induced switching for Ti(1)/Mn₃Sn(12) in addition to Ti(2)/Mn₃Sn(12). From Fig. 10d-e, evident switching is also shown in Ti(1)/Mn₃Sn(12), with comparable switching ratio (~18%) and switching current (~20 mA) to the Ti(2)/Mn₃Sn(12) case. Therefore, decreasing the Ti thickness does not lead to significant increase of switching ratio or decrease of switching current. Furthermore, for Ti(1)/Mn₃Sn(12)/Ti(1) sample with more symmetric structure than Ti/Mn₃Sn, the switching (as displayed in Fig. 10g-h) also shows same polarity, similar level of switching ratio (~21%) and switching current (~20 mA) as the Ti(1,2)/Mn₃Sn(12), despite the existence of both upper and lower Ti/Mn₃Sn interfaces. If the switching is induced by spin current from the interface effect, we would expect nearly zero switching, or at least much smaller switching ratio in Ti(1)/Mn₃Sn(12)/Ti(1). However, the experiment result is very contrary to it. Combined with the switching results in Mn₃Sn/MgO and MgO/Mn₃Sn/MgO, it is suggested that the switching polarity is not related to the interface, and instead, it is the type of seed layer under Mn₃Sn plays an important role in determining the switching polarity in Mn₃Sn.

At last, regarding the different switching polarity in Ti/Mn₃Sn and MgO/Mn₃Sn structures, we believe it is related to the particular crystalline structures of Mn₃Sn, as discussed above in response to Comment #3 of Reviewer 1. Please refer to the relevant paragraphs on Page 7-14 of this rebuttal.

FIG. 10. **a-b**, Field dependence of Hall resistivities of Ti(t_{Ti})/Mn₃Sn(12) with Ti thickness t_{Ti} ranging from 1 nm to 5 nm. **b** is a zoom-in plot of $\rho_H - H_z$ for Ti(3,4,5)/Mn₃Sn(12). **c, f**, Field dependence of Hall resistance of Ti(1)/Mn₃Sn(12) and Ti(1)/Mn₃Sn(12)/Ti(1), respectively. **d-e**, Current dependence of Hall resistance of Ti(1)/Mn₃Sn(12) with an in-plane assistant field H_x of +600 Oe and -600 Oe, respectively. **g-h**, Current dependence of Hall resistance of Ti(1)/Mn₃Sn(12)/Ti(1) with an in-plane assistant field H_x of +600 Oe and -600 Oe, respectively.

In response to the reviewer's comment, we have made the following changes:

s) Page 16-18, line 344-390:

Revised: “As revealed by XRD and magnetic measurements, the crystalline grains in polycrystalline Mn₃Sn may be divided into three groups based on their kagome plane orientation with respect to the

substrate: i) $(11\bar{2}0)$ and $(20\bar{2}0)$, ii) $(10\bar{1}1)$, $(20\bar{2}2)$ and $(20\bar{2}1)$, and iii) (0002) . Grains with (0002) orientation do not contribute to the measured AHE signal and therefore, it cannot be analyzer. In general, when the charge current passes through the analyzer, spin current with both transverse and longitudinal polarization is generated. In the case of configurations shown in Fig. 5b,c, where only spin current with transverse polarization plays a role in switching because the longitudinal spin polarization is perpendicular to the kagome plane. Here, the primed coordinate system is the local coordinate system attached to the polarizer such that x' and y' are always in the kagome plane, which is to be differentiated from the global coordinate system shown in Fig. 5a. In the configuration shown in Fig. 5b, an assistant field H_x is used to align \mathbf{m} of the polarizer along x -direction. In this case the spin current from the polarizer may be written as $j_s = j_{x'x'}^{x'} + j_{x'y'}^{y'}$ (note x and x' are in the same direction in this case). By convention, we use $j_{\beta\gamma}^\alpha$ to denote the spin current flowing in β direction with a polarization along α , induced by a charge current in γ direction. When the spin-polarized current enters the analyzer, spins with transverse polarization $j_{x'y'}^{y'}$, will lead to magnetization switching due to the STT. The switching polarity should depend on the delicate balance of the spin-polarized current from neighbouring grains. On the other hand, in the configuration shown in Fig. 5c, the polarizer and analyzer inter-change their roles, in which STT-based switching is also possible when an assistant field is applied in y -direction (See Supplementary Information S9). However, as the spin polarization in the two cases may be different in amplitude, the ratio of switched regions can differ significantly in the two cases. In actual samples, the polarizer and analyzer are not necessarily always perpendicular to each other as revealed by both XRD and SQUID results. Figure 5d-f show three possible configurations which can lead to both magnetization switching and detectable AHE. Unlike cases in Fig. 5b and 5c, it is difficult to decompose j_s into spin current with longitudinal and transverse polarizations in the local coordinates as the kagome planes of the polarizer and analyzer do not form a right-angle. In these cases, depending on the exact kagome plane orientation of the polarizer and analyzer, spin currents with both transverse and longitudinal spin polarization may cause the switching of the magnetization. In the current measurement configuration, phenomenologically the AHE signal is proportional to the vertical component of the net magnetization in the kagome plane. Therefore, the configurations shown in Fig. 5d-f would lead to switching with a polarity opposite to that of Fig. 5b and 5c.

As discussed earlier, the texture of polycrystalline Mn_3Sn is well reflected in its M - H loop. From the decomposition of M - H curves as shown in Fig. 1d, we find that in the case of $\text{Ta}/\text{MgO}(t_{\text{MgO}})/\text{Mn}_3\text{Sn}$, the contribution from group i) grains increases with increasing the MgO thickness, indicating that MgO seed layer facilitates the formation of group i) grains and the switching in MgO/ Mn_3Sn might

be dominated by the configuration in Fig. 5b and 5c. In addition, as shown in Supplementary Information S2, the $M-H$ loop for Ti(2)/Mn₃Sn(60) is very similar to that of Ta/MgO(2)/Mn₃Sn, at which sign reversal has not occurred yet, *i.e.*, the switching polarity is the same for Ta/Mn₃Sn, Ti/Mn₃Sn, and Ta/MgO(2)/Mn₃Sn. The XRD profiles and temperature dependent AHE of Mn₃Sn in different structures further suggest that there are mostly group ii) and iii) grains in Ti/Mn₃Sn (Supplementary Information S1 and S7). Therefore, the switching configurations in Fig. 5d-f may become dominant in Ti/Mn₃Sn. At this stage, however, a quantitative analysis of the switching polarity would be difficult considering the random distribution of the crystalline orientation of the grains.”

t) Page 21-22, line 460-474:

Added: “In addition, the interface-related effect such as Rashba effect is not likely to play an important role here, as in samples of Ta/MgO/Mn₃Sn/MgO/Ta and MgO/Mn₃Sn/MgO/Ta, we always kept the bottom and upper MgO layer the same thickness to minimize structural asymmetry. If there is any significant Rashba effect between MgO and Mn₃Sn interface, the effects from upper and lower interfaces will be largely cancelled out. The interface-related effect also fails to account for the negligible switching in substrate/Mn₃Sn/MgO/Ta, as structural asymmetry is more prominent in this structure. Furthermore, previous study has excluded the interface effect in current-induced switching of Mn₃Sn by inserting a Cu layer between Mn₃Sn and heavy metal layer. Previous studies also suggest negligible charge-spin conversion at Ti/ferromagnetic interface. In addition, as shown in Supplementary Information S8, the same switching polarity and similar level of switching ratio in Ti/Mn₃Sn/Ti symmetric structure with Ti/Mn₃Sn further attests that the switching is not induced by interface effect. After the submission of this manuscript, several theoretical and experimental works have been published which support the spin current generation and grain-based switching mechanism discussed in this study.”

u) Added current-induced switching of Ti(1)/Mn₃Sn(12) and Ti(1)/Mn₃Sn(12)/Ti(1) in Supplementary Information S8.

Comment 2: “An electric current passing through a grain (Polarizer) contributes to the switching of a neighboring grain (Analyzer) only when the specific relative crystalline orientations between the two grains are realized (there are two such configurations out of the twenty-four possible ones, as discussed by the authors). And in any case, Polarizer grains themselves do not change their magnetizations under the current. So, there must be way more regions in a sample that are not affected by the current than those change their magnetizations in a way that the change is reflected in the MOKE signal. Considering these things, I wouldn't expect the proposed inter-grain STT leads

to such an appreciable contrast in the MOKE signal as in Fig. 3. Can the authors make some comment on this point?”

Response: Thank the reviewer for the insightful comment. MOKE is unable to “see” switching of individual grains given its sub-micron level spatial resolution. If the switching only stops at nearest neighbouring grains, in principle, one would only be able to obtain a grey level image with slight change in brightness. The clusters observed in the MOKE image are believed to result from “chain” switching like events due to interchangeable role of polarizer/analyzer once the switching is initiated. The size of the switched cluster depends on the grain configurations in the local area. This may explain the difference in MOKE images of Ta(2)/MgO(2)/Mn₃Sn(12) and Ta(2)/MgO(4)/Mn₃Sn(12) after application of a longitudinal current (Fig. 11a and 11b). The switching of the former is dominated by spin current from Ta layer, which is uniformly acting on Mn₃Sn layer, while the switching of the latter is induced by inter-grain STT as discussed above. The other possibility is that the large switched-area in the MOKE image may actually consist of densely packed switched grains, and the grey-scale contrast simply corresponds to the density of switched grains, considering the limited spatial resolution of MOKE. After we have submitted the manuscript, there is a paper recently published reporting systematic studies of magnetization switching in Mn₃Sn/Pt bilayers using NV magnetometry. As detailed information such as lift height and number of NV centers is not known, it is difficult to estimate the spatial resolution of the specific apparatus used, but it is presumably better than the MOKE microscopy we used. As shown in Fig. 11c, a clear granular-like switching is also observed in Mn₃Sn/Pt. Based on the scale bar given in the figure (2.5 μm), the size of switched area ranges from sub-micron to a few microns. Some of the inter-connected and densely packed areas may appear as a single patch in the MOKE image. Therefore, the switching ratio estimated from the MOKE image may not have a one-to-one correspondence with the AHE result, *i.e.*, 14% for Ta(2)/MgO(4)/Mn₃Sn(12).

FIG. 11. a-b, The MOKE images of Ta(2)/MgO(2)/Mn₃Sn(12) and Ta(2)/MgO(4)/Mn₃Sn(12) after application of a longitudinal current, respectively (figures taken from manuscript Fig. 3). **c,** Image of the SOT-induced

change of the out-of-plane stray field ΔB_z from $\text{Mn}_3\text{Sn}/\text{Pt}$. Blue areas are switched parts of Mn_3Sn after current injection (figure taken from (G. Q. Yan, et. al., Adv. Mater. 2200327 (2022))).

In response to the reviewer's comment, we have made the following changes:

v) Page 12, line 255-257:

Added: "It should be noted that it is difficult to estimate the switching ratio from the MOKE image due to the limited spatial resolution of MOKE imaging."

Comment 3: "Also, if it is the case that the inter-grain STT indeed plays the essential role in the switching, the authors should be able to get some insights into the crystalline orientations of each grain contained in the sample by comparing the MOKE results before and after the current pulse applications, and discuss if the current-induced changes in the grains are consistent with the inter-grain-STT scenario."

Response: We understand where the reviewer's question is coming from. Unfortunately, this is quite challenging and not possible due to limited spatial resolution of MOKE. The grain size is in the range of 10-40 nm (from TEM images shown in Fig. 2 of the rebuttal), which is more than 10 times smaller than the spatial resolution of the MOKE we used. As mentioned above, the large clusters in the MOKE image may originate from densely packed grains of which majority of them has been switched. They appear as a large patch with same brightness because of the low spatial resolution of the MOKE microscope. Therefore, it is impossible to relate crystalline orientations of individual grains to the MOKE image, though it would be wonderful if we could do so. A combination of *in-situ* TEM and spin-dependent STM may be able to achieve the goal, but it is certainly out of the scope of this study due to unavailability of the facility. Nevertheless, we would like to thank the reviewer for the good suggestion.

In response to the reviewer's comment, we have made the following changes:

w) Page 19, line 406-412:

Added: "As mentioned above, from MOKE images for $\text{Ta}(2)/\text{MgO}(4)/\text{Mn}_3\text{Sn}(12)$ in Fig. 4c-d, we can see that the switched areas are discrete and more non-uniformly distributed than $\text{Ta}(2)/\text{MgO}(2)/\text{Mn}_3\text{Sn}(12)$ case. This is consistent with the inter-grain STT switching scenario as the switching occurs only when a pair of neighbouring grains have specific configurations as one shown in Fig. 5b-f. However, it is not quite possible to determine the exact crystalline orientations of individual grains from the MOKE images due to limited spatial resolution of MOKE imaging."

x) Page 20, line 429-431:

Added: “A combination of *in-situ* TEM and spin-dependent scanning tunnelling microscopy may be able to obtain more insights, which is out of the scope of this study.”

Comment 4: “As several groups have already reported experimental demonstrations of current-induced switching/rotation of the magnetizations in Mn₃Sn thin films, the (almost only) major advance of this work is the claimed elucidation of the new physical switching mechanism, that is, the inter-grain STT in polycrystalline thin films. Therefore, I believe that the authors must provide more clear-cut/direct experimental evidence that the inter-grain STT is indeed playing the dominant role in the observed switching.”

Response: The previously reported experimental demonstrations of current-induced switching in Mn₃Sn are still based on the scenario of conventional SOT-induced switching in heavy metal/ferromagnet, in which Mn₃Sn is treated similarly as a ferromagnet. Spin current generation inside Mn₃Sn is completely ignored, which recent theoretical studies showed that may play a crucial role in self-switching of spin chirality in Mn₃Sn or Mn₃Sn-based tunnel junctions (J. Železný, et. al., Phys. Rev. Lett. 119, 187204 (2017); D. Go et al., arXiv:2201.11476; J. Dong et al., arXiv:2112.06568). Unlike polycrystalline ferromagnets in which magnetizations can get aligned across crystalline grains, the spin structure of Mn₃Sn is largely “contained” inside individual grains, in other words, each crystalline grain can be treated approximately as a magnetic grain as manifested in both the MOKE imaging results of this work and the NV magnetometry results mentioned above. Therefore, naturally one would expect a self-generated and grain-specific spin current inside polycrystalline Mn₃Sn. Such kind of self-generated spin current with non-uniform polarization direction is bound to cause current-induced switching of part of the grains. Therefore, we believe it should be treated on the equal footing with externally generated spin current.

To support the inter-grain STT switching scenario, we performed TEM imaging to view the detailed layer structure and crystalline property in the thin films for self-induced STT switching. From the cross-sectional TEM images shown in Fig. 2a-b in the rebuttal, we can see the polycrystalline structure of deposited film, with the grain size differing from 10 nm to 40 nm. Fig. 12a displays the high-resolution (HR) TEM image for Ta/MgO(4)/Mn₃Sn, from which the structure for each layer is clearly seen. Despite the post-annealing at 450°C, there is no intermixing of Mn₃Sn with Ta layer due to the MgO insertion layer, and the crystalline texture for Mn₃Sn layer is visible. By performing fast Fourier transform for the Mn₃Sn layer in Fig. 12a, we obtained the corresponding electron diffraction pattern. In the diffraction pattern in Fig. 12b, we marked each point from close

to far from the centre as ①-⑤. Based on the distances of these points to the centre, we further obtained the interplane spacings and the corresponding crystalline planes, as summarized in Table I. The crystalline planes determined from TEM images is consistent with the XRD measurement. Multiple planes are detected in such small imaging area with width of around 37 nm, indicating the existence of grains with different crystalline planes in the lateral direction of the sample. In addition, the twin spots observed in Fig. 12b at ①②③⑤ further confirm the grains existence and imply that the grain boundaries are in the form of twin boundaries (E. Sakedai, et. al., *Int. J. Mater. Res.* 101, 736 (2010); M. Martinez, et. al., *Mater. Charact.* 134, 76 (2017); P. Uttam, et. al., *Mater. Des.* 192, 108752 (2020)). As mentioned above, a recent study also finds that there exist multiple grains in polycrystalline Mn_3Sn via NV magnetometry imaging. Individual grain is like a domain with specific crystalline and magnetic orientation. The magnetic state in each grain switches independently due to the grain boundary ((G. Q. Yan, et. al., *Adv. Mater.* 2200327 (2022))).

Regarding the inter-grain STT, the spin-polarized current in Mn_3Sn has already been theoretically reported. Therefore, to prove that the spin-polarized current can exert torque to the neighbouring grain, the best way is to model the STT switching in a system with two neighbouring Mn_3Sn grains. After we have submitted the manuscript, several theoretical papers have been published which predicts spin-polarized current induced switching in non-collinear AFM-based magnetic junction (S. Ghosh, et. al., *Phys. Rev. Lett.* 128, 2097702 (2022); J. Dong et al., *Phys. Rev. Lett.* 128, 197201 (2022)), as well as self-induced switching of chiral magnetic textures in non-collinear AFM (D. Go, et. al., arXiv:2201.11476). Though the detailed plane configuration and switching mechanism in these studies are different with each other, they do point out the possibility of realizing self-induced switching in Mn_3Sn with the bulk spin current generated inside itself. Therefore, we believe the combination of results presented in this study and experimental/theoretical papers published recently by others provides a convincing scenario about the inter-grain STT based switching in polycrystalline Mn_3Sn films without an adjacent heavy metal. When the heavy metal is present, both externally and internally generated spin current should be treated on equally footing when dealing with current induced switching.

FIG. 12. a, High resolution Cross-sectional TEM images of Ta/MgO(4)/Mn₃Sn. **b**, Election diffraction pattern of the Mn₃Sn layer in **c**.

Table I. The crystalline planes determined from the electron diffraction pattern in Fig. 8b.

Number	Interplane spacing (Å)	Crystalline plane
①	3.40	(10 $\bar{1}$ 1)
②	3.26	(10 $\bar{1}$ 1)
③	2.89	(11 $\bar{2}$ 0)
④	2.30	(0002)
⑤	1.70	(20 $\bar{2}$ 2)

In response to the reviewer’s comment, we have made the following changes:

y) Page 19-20, line 412-429:

Added: “To support this scenario, we further examined the crystalline structure using transmission electron microscopy (TEM). From the cross-sectional TEM images in Fig. 6a-b, the Mn₃Sn grain size differs from 10 nm to 40 nm. Figure 6c displays the high-resolution (HR) cross-sectional TEM image for Ta/MgO(4)/Mn₃Sn, from which the structure for each layer is clearly shown. Despite the post-annealing at 450°C, there is no apparent intermixing of Mn₃Sn with Ta layer due to the MgO insertion layer, and the crystalline texture for Mn₃Sn layer is visible. By performing fast Fourier transform for the Mn₃Sn layer in Fig. 6c, we obtained the corresponding electron diffraction pattern

as shown in Fig. 6d. Based on the distances of the spots to the centre, we further obtained the interplane spacings and the corresponding crystalline planes, as marked near each spot. The crystalline planes determined from TEM images is consistent with the XRD measurement. Multiple planes are detected in the small imaging area with width of around 37 nm, indicating the existence of grains with different crystalline planes in the lateral direction of the sample. In addition, the twin spots observed in Fig. 6d further confirm the grains existence and imply that the existence of grain boundaries in the form of twin boundaries. A recent study also finds that there exist multiple grains in polycrystalline Mn_3Sn and individual grain is like a domain with specific crystalline and magnetic orientation. The magnetic state in each grain switches independently due to the grain boundary.”

z) Added TEM images in Fig. 6.

Other changes made:

1. Revised the typo (2021) to $(20\bar{2}1)$ in the text of Fig. 1b.

2. Page 7, line 160:

Revised subheading to: “**Current-induced switching in Mn_3Sn** ”.

2. Page 13, line 279-282:

Original: “The aforementioned results suggest that both the external spin current from the Ta layer and the self-generated spin current from the neighbouring grains contribute to spin dynamics in Mn_3Sn .”

Revised: “**The aforementioned results suggest that in addition to the external spin current from the Ta layer, there is another spin current source contributing to the spin dynamics of a particular grain in polycrystalline Mn_3Sn , which is from neighbouring grains with misaligned spin sublattices.**”

3. Updated the figure captions for the revised figures.

4. Moved assistant field angle dependence of current-induced switching from the original main text to Supplementary Information S9.

5. Revised Supplementary Information according to the change in manuscript.

With these changes, we hope that the manuscript is now acceptable for publishing in your journal. We look forward to your positive response.

Yours sincerely,

Dr. Hang Xie

Department of Electrical and Computer Engineering,

National University of Singapore,

Tel: (65) 65162139; E-mail: elehang@nus.edu.sg

Reviewers' Comments:

Reviewer #1:

Remarks to the Author:

The authors address the previous comments and questions well. The quality of the work has been greatly improved. There are some other suggestions and questions based on the response.

1. As the authors said, the inter-grain spin-transfer torque has advantages mostly for the applications which requires the output to be nonlinear function of the input. The authors could add it into introduction part or even abstract, to reflect the importance.
2. The authors demonstrate the correlation of switching ratio with AHE remanence in the revised letter and compared the switching ratio of SOT between that of IGSTT. The results show that the switching ratio of IGSTT is much smaller than that of SOT, and this disadvantage originates from the fundamental physical mechanism of IGSTT. I think this will greatly hamper the application and development of IGSTT in noncollinear antiferromagnets. Do the author have any methods or ideas to conquer this disadvantage? I do not think growing Mn₃Sn with only 2 grains is a feasible method, although the authors said in the revised letter.
3. The format of the references should keep the same style.

Reviewer #2:

Remarks to the Author:

I appreciate the authors for their extensive explanations on the work and the revisions made in the manuscript.

I'm still not convinced that the authors' claim of the inter-grain STT being the major mechanism behind their observations has been clearly confirmed. That being said, the topic is timely and the manuscript seems to contain some potentially important results that should be recognized and discussed widely in the community, and thus worths publication.

I would like the authors to tone down their claim a little bit in the manuscript (especially introduction and conclusions). In my eyes, what the authors can assert with some confident, from their experimental data, is; the spin Hall current injection and the Rashba effect are both very unlikely to play a major role in the present system. Then, I get that it is quite reasonable to suspect the inter-grain STT mechanism as a cause of the observed switching. But, I don't believe that the displayed data is providing a direct evidence that exclusively supports that scenario. And, the fact that the switching polarity depends on the nonmagnetic layers attached onto Mn₃Sn, in my eyes, is fundamentally contradicting the notion of "self-induced" torque, and probably should be taken more seriously. It might actually be implying some important physical effects caused by remaining structural inversion asymmetry.

The current manuscript sounds more like that the authors actually identified the inter-layer STT as THE microscopic origin of the observation, other possible effects being unambiguously excluded. I don't think that is the case. With a little more down-to-earth presentations of the experimental observations, I think the manuscript deserves to be published with no more delay so that the community can widely discuss the results.

We would like to thank the reviewers again for taking time to review the revised manuscript and providing insightful comments. Below we include a summary of changes in the 2nd revision and point-to-point responses to reviewers' comments:

A) Summary of changes (highlighted in red in the revised manuscript)

- 1) Added the advantage of IGSTT in applications requiring nonlinear output in introduction.
- 2) Discussed more about how to increase the switching ratio.
- 3) Checked and revised reference details and style.
- 4) Toned down the claim in introduction and summary.

B) Response to comments from Reviewer #1

Comment 1: “As the authors said, the inter-grain spin-transfer torque has advantages mostly for the applications which requires the output to be nonlinear function of the input. The authors could add it into introduction part or even abstract, to reflect the importance.”

Response: We would like to thank the reviewer for the suggestion. We have added this point into the end of introduction. We did not add into the abstract due to the length limit.

As a response to the reviewer's comment, we have made the following change:

a) Page 3-4, line 83-88:

Original: “Furthermore, the removal of the thick HM layer that was used in previous studies significantly increases the AHE signal by one-two orders of magnitude, making it more appealing for applications.”

Revised: “Furthermore, the removal of the thick HM layer that was used in previous studies significantly increases the AHE signal by more than one order of magnitude, making it more appealing for energy-efficient device applications. Accurate control of grain size and its distribution may provide an effective knob to tune the switching characteristics that is more suitable for emerging applications such as neuromorphic computing.”

Comment 2: “The authors demonstrate the correlation of switching ratio with AHE remanence in the revised letter and compared the switching ratio of SOT between that of IGSTT. The results show that the switching ratio of IGSTT is much smaller than that of SOT, and this disadvantage originates from the fundamental physical mechanism of IGSTT. I think this will greatly hamper the application

and development of IGSTT in noncollinear antiferromagnets. Do the author have any methods or ideas to conquer this disadvantage? I do not think growing Mn_3Sn with only 2 grains is a feasible method, although the authors said in the revised letter.”

Response: Thank the reviewer for the comment. The maximum switching ratio observed from the IGSTT-induced switching is indeed smaller than that of SOT-induced switching in Mn_3Sn , which however may not render it less promising than the SOT-based Mn_3Sn switching in device applications. As mentioned in the manuscript, the removal of a relatively thick HM layer used for external spin current generation greatly increases the AHE signal due to less current shunting. In Table I below, we compare our results with those reported in HM/ Mn_3Sn bilayers. Although the switching ratio of the IGSTT-based single layer device is relatively small, the AHE resistance change caused by magnetization switching is much larger than those achieved in HM/ Mn_3Sn bilayers. The latest issue of Nature published a work from the Univ. of Tokyo group, which demonstrated 100% perpendicular switching of chiral AFM order in epitaxial W/ Mn_3Sn thin films with appropriate strain engineering (last row of Table I) and is certainly a significant step forward towards device applications. While the device may not be optimized, the AHE amplitude reported in that work is still much smaller than the value obtained in our single layer structure. Therefore, though the switching ratio of IGSTT-induced switching in single layer Mn_3Sn is smaller than SOT-based switching in HM/ Mn_3Sn bilayers, the readout signal amplitude turned out to be otherwise, at least, at the current stage of development of both types of devices.

We understand the reviewer’s concerns, but in our perspective further investigations should be carried out in both directions given the limited number of works reported so far. For the HM/ Mn_3Sn bilayers, the key is to find a way to increase the readout amplitude. In order to suppress the current shunting effect, one may eventually need to resort to a three terminal device to increase the output signal to a practically useful level, e.g., to readout the magnetic state using a vertical AFM tunnel junction. Although Mn_3Sn -based tunnel junction has been proposed theoretically, to the best of our knowledge, it has yet to be demonstrated experimentally. On the other hand, for the IGSTT-based switching, there are two possible directions. One is to achieve desirable switching characteristics in large-size samples through grain size/distribution engineering for specific applications, and the other is to realize bi-grain or few-grain structures in small-size samples. The latter will lead to further insights of IGSTT-based switching and enhancement of switching efficiency, or eventually the realization of lateral tunnel junctions. Once a lateral tunnel junction is realized, it is possible to have a two-terminal AFM-based device with large output signal.

Table I. Summary of switching ratio and current-induced AHE change amplitude of different Mn₃Sn-based structures in reported literature and this work.

Structure	Switching ratio	Current-induced AHE change amplitude (Ω)	Refs.
Ti(1)/Mn ₃ Sn(12)/Ti(1)	21%	0.5	This work
Ru(2)/Mn ₃ Sn(40)/Pt(7.2)	30%	0.0225	Nature 580, 608 (2020)
W(10)/Ta(1)/Mn ₃ Sn(30)/Pt(4)	43%	0.01	Nat. Mater. 10, 1038 (2021)
TaN(3)/W(8)/Mn ₃ Sn(40)	40%	0.02	Sci. Adv. 8, eabo5930 (2022)
W(7)/Mn ₃ Sn(30)	100%	0.09	Nature 607, 474 (2022)

In response to the reviewer’s comment, we have made the following changes:

b) Page 21, line 452-456:

Original: “Therefore, further studies may be required to produce well textured films with desirable grain orientation for current-induced switching.”

Revised: “Therefore, further studies may be required to produce well textured films with desirable grain orientation/size for current-induced switching. The active switching region may be reduced to contain only a few or ultimately two grains, thereby allowing to gain further insights of the switching mechanism and enhance the switching efficiency.”

Comment 3: “The format of the references should keep the same style.”

Response: Thank the reviewer for pointing this out. We have checked the reference styles and details thoroughly and revised accordingly where necessary.

C) Response to comments from Reviewer #2

Comment 1: “I would like the authors to tone down their claim a little bit in the manuscript (especially introduction and conclusions). In my eyes, what the authors can assert with some confident, from their experimental data, is; the spin Hall current injection and the Rashba effect are both very unlikely to play a major role in the present system. Then, I get that it is quite reasonable to suspect the inter-grain STT mechanism as a cause of the observed switching. But, I don't believe that

the displayed data is providing a direct evidence that exclusively supports that scenario. And, the fact that the switching polarity depends on the nonmagnetic layers attached onto Mn₃Sn, in my eyes, is fundamentally contradicting the notion of "self-induced" torque, and probably should be taken more seriously. It might actually be implying some important physical effects caused by remaining structural inversion asymmetry.

The current manuscript sounds more like that the authors actually identified the inter-layer STT as THE microscopic origin of the observation, other possible effects being unambiguously excluded. I don't think that is the case. With a little more down-to-earth presentations of the experimental observations, I think the manuscript deserves to be published with no more delay so that the community can widely discuss the results.”

Response: We thank the reviewer for the comment and suggestion. The proposed IGSTT mechanism is based on the correlation among structural, magnetic, magneto-optical, and electrical transport measurement results, and the unique spin structure of non-collinear AFM. We believe both the SHE and Rashba effect are not playing a role based on the apparent structural symmetry of the device. However, we also agree with the reviewer that we cannot completely exclude the possibility that there might be a hidden structural inversion asymmetry which is not apparent from the pre-designed device structure, especially at the two interfaces. Therefore, we have revised the manuscript to tone down the claims.

In response to the reviewer’s comment, we have made the following changes:

c) Page 1, line 18-21:

Original: “The inter-grain spin-transfer torque induced by self-generated spin-polarized current explains the experimental observations. Our findings demonstrate the crucial role of self-generated spin current in Mn₃Sn and provide an alternative pathway for electrical manipulation of non-collinear antiferromagnetic state without resorting to the conventional bilayer structure.”

Revised: “The inter-grain spin-transfer torque induced by spin-polarized current explains the experimental observations. Our findings provide an alternative pathway for electrical manipulation of non-collinear antiferromagnetic state without resorting to the conventional bilayer structure.”

d) Page 22-23, line 489-494:

Original: “We hope our results will stimulate more interest in IGSTT, in particular, in non-collinear antiferromagnets.”

Revised: “At last, it is worth mentioning that we do not completely rule out other possible mechanisms such as the presence of hidden structural asymmetry that is not apparent in the as-deposit layer structure. In addition, the joule heating effect has also been reported recently to play a crucial role in the switching behavior of Mn_3Sn . We hope all these combined may stimulate more investigations in current-induced switching of non-collinear antiferromagnets.”

Other changes made:

1. Checked through the manuscript and revised the typos.

With these changes, we hope that the manuscript is now acceptable for publishing in your journal. We look forward to your positive response.

Reviewers' Comments:

Reviewer #1:

Remarks to the Author:

The authors address my previous comments and questions very well. I recommend its publication as it is.